# Concerted action of kinesins KIF5B and KIF13B promotes efficient secretory vesicle transport to microtubule plus ends

Andrea Serra-Marques[1†‡], Maud Martin[1†§], Eugene A Katrukha[1], Ilya Grigoriev[1], Cathelijn AE Peeters[1], Qingyang Liu[1], Peter Jan Hooikaas[1], Yao Yao[2], Veronika Solianova[1], Ihor Smal[2#], Lotte B Pedersen[3], Erik Meijering[2¶], Lukas C Kapitein[1], Anna Akhmanova[1*]

[1]Cell Biology, Neurobiology and Biophysics, Department of Biology, Faculty of Science, Utrecht University, Utrecht, Netherlands; [2]Departments of Medical Informatics and Radiology, Biomedical Imaging Group Rotterdam, Erasmus University Medical Center, Rotterdam, Netherlands; [3]Department of Biology, Section of Cell Biology and Physiology, the August Krogh Building, University of Copenhagen, Copenhagen, Denmark

*For correspondence:
a.akhmanova@uu.nl

[†]These authors contributed equally to this work

Present address: [‡]Department of Bioengineering and Therapeutic Sciences, University of California, San Francisco, San Francisco, United States; [§]Laboratory of Neurovascular Signaling, Department of Molecular Biology, Universite ́ libre de Bruxelles (ULB), Gosselies, Belgium; [#]Theme of Biomedical Sciences, Erasmus University Medical Center, Rotterdam, Netherlands; [¶]Faculty of Engineering, the University of New South Wales, Sydney, Australia

**Abstract** Intracellular transport relies on multiple kinesins, but it is poorly understood which kinesins are present on particular cargos, what their contributions are and whether they act simultaneously on the same cargo. Here, we show that Rab6-positive secretory vesicles are transported from the Golgi apparatus to the cell periphery by kinesin-1 KIF5B and kinesin-3 KIF13B, which determine the location of secretion events. KIF5B plays a dominant role, whereas KIF13B helps Rab6 vesicles to reach freshly polymerized microtubule ends, to which KIF5B binds poorly, likely because its cofactors, MAP7-family proteins, are slow in populating these ends. Sub-pixel localization demonstrated that during microtubule plus-end directed transport, both kinesins localize to the vesicle front and can be engaged on the same vesicle. When vesicles reverse direction, KIF13B relocates to the middle of the vesicle, while KIF5B shifts to the back, suggesting that KIF5B but not KIF13B undergoes a tug-of-war with a minus-end directed motor.

## Introduction

Intracellular transport is driven by the collective activity of multiple motors. Transport toward microtubule (MT) plus ends depends on several kinesin families, whereas minus-end directed movement in animal cells is mostly driven by cytoplasmic dynein (*Akhmanova and Hammer, 2010*; *Hirokawa and Tanaka, 2015*). Different types of kinesins can participate in transporting the same cargo, but for many cellular organelles it is currently unclear which motors contribute to motility, what their specific roles are and whether different types of motors can be engaged in transport simultaneously. Furthermore, a common feature of many cellular cargos is their bidirectional transport along MTs, and it is well established that both kinesins and dynein can operate on the same cargo. Motors of opposite polarity can engage in a tug-of-war, be switched on and off in a coordinated fashion or even depend on each other for motility (*Gross, 2004*; *Hancock, 2014*; *Welte, 2004*). Distinguishing these mechanisms requires the detection of motor activity on cellular cargo, which in many situations proved to be highly challenging.

Carriers of constitutive secretion represent a convenient and functionally important cellular model to study intracellular transport. Members of the plus-end directed kinesin-1 family (KIF5A/B/C) and kinesin-3 (KIF13A, KIF13B) have been implicated in the transport of Golgi-derived carriers to the cell surface in different cell types (*Astanina and Jacob, 2010*; *Burgo et al., 2012*; *Jaulin et al., 2007*;

*Nakagawa et al., 2000*; *Yamada et al., 2014*). In both constitutive and inducible secretion assays in mammalian cells, exocytotic carriers positive for different exocytotic cargos, such as NPY or temperature sensitive VSV-G, can be readily labeled with the small GTPases Rab6 and Rab8 (*Grigoriev et al., 2007*; *Grigoriev et al., 2011*; *Jasmin et al., 1992*; *Miserey-Lenkei et al., 2010*; *Wakana et al., 2012*). Recent work conclusively demonstrated that the majority of post-Golgi carriers, irrespective of the marker used, are positive for Rab6, which appears to be a general regulator of post-Golgi secretion and thus represents an excellent marker for secretory vesicles (*Fourriere et al., 2019*). It was further shown that cytoplasmic dynein drives transport of Rab6-positive membranes to MT minus ends, whereas kinesin-1 KIF5B and the kinesin-3 family members KIF1B and KIF1C have been implicated in the plus-end directed motility (*Grigoriev et al., 2007*; *Lee et al., 2015*; *Matanis et al., 2002*; *Schlager et al., 2010*; *Schlager et al., 2014*; *Short et al., 2002*; *Wanschers et al., 2008*). Moreover, KIF1C was shown to interact with Rab6 directly (*Lee et al., 2015*). However, siRNA-mediated co-depletion of KIF5B, KIF1B and KIF1C was not sufficient to block MT plus-end directed movement of Rab6 vesicles in HeLa cells (*Schlager et al., 2014*), and it is thus possible that additional kinesins are involved. For example, it has been proposed that the Golgi-derived carriers named CARTS (carriers of the TGN to the cell surface), which are labeled with Rab6, Rab8 and the protein cargo pancreatic adenocarcinoma up-regulated factor (PAUF) are driven by the mitotic kinesin-5 family member KIF11/Eg5 (*Ferenz et al., 2010*; *Wakana et al., 2012*; *Wakana et al., 2013*), but the exact contribution of this motor requires further clarification.

Another important question concerns the specific roles of different kinesins on the same cargo. An increasing body of evidence suggests that different kinesins can preferentially bind to specific MT tracks, and the presence of different kinesins on the same cargo might thus help these cargos to navigate heterogeneous MT networks. For example, kinesin-1 shows a strong preference for more stable MT populations enriched in acetylated and detyrosinated tubulin, whereas kinesin-3 prefers dynamic, tyrosinated MTs (*Cai et al., 2009*; *Guardia et al., 2016*; *Konishi and Setou, 2009*; *Liao and Gundersen, 1998*; *Lipka et al., 2016*; *Tas et al., 2017*). These kinesins also require different MT-associated proteins (MAPs) for their activity: kinesin-1 critically depends on MAP7 family proteins whereas the activity of kinesin-3 is stimulated by doublecortin and doublecortin-like kinase (*Chaudhary et al., 2019*; *Hooikaas et al., 2019*; *Lipka et al., 2016*; *Liu et al., 2012*; *Métivier et al., 2019*; *Metzger et al., 2012*; *Monroy et al., 2018*; *Pan et al., 2019*). Whether and how the MAP-kinesin cross-talk contributes to motor selectivity is currently unknown.

In addition to kinesin preferences for specific tracks, other differences in the properties of the motors can affect their collective behaviors. In vitro and cellular studies have shown that kinesin-3 motors are faster than kinesin-1, but detach more easily from MTs, and therefore, when the two motors are combined in vitro, kinesin-1 predominates (*Arpağ et al., 2019*; *Arpağ et al., 2014*; *Norris et al., 2014*). In cells, the situation is more complex, because the presence of MAPs and post-translational tubulin modifications can affect MT-motor interaction and thus determine which motor will dominate (*Gumy et al., 2017*; *Norris et al., 2014*). It was also proposed that in situations where kinesin-1 is dominant, kinesin-3 can enhance overall transport efficiency by preventing cargo detachment from MTs and helping to navigate around obstacles (*Arpağ et al., 2019*; *Norris et al., 2014*). Importantly, much of what is known about motor preferences for specific tracks and the details of their individual and collective properties is based on the observation of kinesin rigor mutants or kinesin fragments, either unloaded or recruited to artificial cargo (*Arpağ et al., 2019*; *Arpağ et al., 2014*; *Guardia et al., 2016*; *Kapitein et al., 2010a*; *Norris et al., 2014*; *Tas et al., 2017*), whereas much less is known about the behavior of motors attached to endogenous cargo by their natural adaptors.

Here, by using gene knockout and rescue experiments, we dissected the composition and behavior of the motor machinery responsible for the transport of secretory vesicles from the Golgi complex to the cell surface. We showed that the transport of Rab6/PAUF vesicles to the cell periphery relies on kinesin-1 KIF5B and kinesin-3 KIF13B. In the absence of both kinesins, the efficiency of secretion was not perturbed, but exocytotic vesicles fused with the plasma membrane close to the Golgi and not at the cell periphery. KIF5B plays a dominant role in the secretory vesicle transport, and this transport is also strongly affected by the depletion of its co-factors, MAP7 family members. MAP7 proteins are more abundant on perinuclear MTs, likely because freshly polymerized MT ends, which are enriched at the cell periphery, are populated by these MAPs with some delay. Accordingly, KIF5B shows less activity on newly grown MT segments compared to the older MT lattices. In

contrast, KIF13B can reach newly grown MT ends efficiently and contributes to the transport of Rab6 vesicles to these ends.

Rab6 vesicle transport in kinesin knockout cells could be rescued by fluorescently tagged versions of KIF5B and KIF13B. High-resolution imaging of these kinesins on moving vesicles showed that when only one type of kinesin was present, it was enriched at the front of a vesicle moving to the cell periphery. When the two kinesins were present on a vesicle simultaneously, the faster KIF13B was located in front of the slower KIF5B. When vesicles reversed direction, KIF13B re-located to the vesicle center, whereas KIF5B was positioned at the back, suggesting that KIF5B but not KIF13B undergoes some mechanical competition with dynein. Our work thus provides insight into the spatial regulation and function of multimotor assemblies during bidirectional cargo transport.

## Results

### The Kinesin-3 KIF13B associates with Rab6-positive secretory vesicles

To image secretory carriers, we used the small GTPase Rab6, which was shown to be a highly robust and ubiquitous marker of exocytotic vesicles (*Fourriere et al., 2019*; *Grigoriev et al., 2007*). In non-neuronal cells, Rab6 is represented by the Rab6A and Rab6A' isoforms that will be collectively called Rab6; in this study, the Rab6A isoform was used in all live imaging experiments. While testing colocalization of Rab6 vesicles with different kinesins, we found that KIF13B, as well as its motorless tail region, were abundantly present on Rab6-positive vesicles in both fixed and live HeLa cells (*Figure 1A–D*). In contrast, its closest homologue KIF13A displayed little binding to Rab6 vesicles (*Figure 1C*, *Figure 1—figure supplement 1A*), in line with previous work showing that KIF13A specifically binds to recycling endosomes by associating with the Rab11 family members Rab11A, Rab11B and Rab25 (*Delevoye et al., 2014*).

The kinesin-5 Eg5/KIF11 has been previously implicated in the transport of the Golgi-derived carriers named CARTS, which are also positive for Rab6 and Rab8 (*Wakana et al., 2012*; *Wakana et al., 2013*). We analyzed the colocalization between the CARTS cargo protein, PAUF-mRFP and Rab6 and found that they indeed colocalized both in fixed cells stained for endogenous Rab6 (*Figure 1—figure supplement 1B*), and also in live cells (*Figure 1E*). Only a sub-population of Rab6-positive vesicles contained PAUF (~60% and 50% in fixed and live cells, respectively) (*Figure 1—figure supplement 1B*). This suggests that Rab6 vesicles may serve multiple exocytotic routes, with PAUF utilizing one of these routes. In line with this view, in secretion assays, less than 20% of PAUF vesicles contained the classic secretion marker, the temperature-sensitive VSV-G (*Wakana et al., 2012*), while a high degree of colocalization of VSV-G and Rab6 has been observed (*Grigoriev et al., 2007*). Even though we were able to confirm that PAUF-positive carriers are labeled with Rab6, we could not detect any colocalization between Rab6 and GFP-Eg5, both in fixed and in live cells (*Figure 1—figure supplement 1C,D*), in contrast with a previous report (*Wakana et al., 2013*). However, PAUF colocalized with KIF13B, as 50.7 ± 22.3% of PAUF-mRFP-positive vesicles were labeled with GFP-KIF13B (determined from 10 cells), and kymograph analysis of PAUF/Rab6-positive vesicles clearly showed that they move together (*Figure 1F*). These results support our observation on KIF13B/Rab6-vesicle colocalization and show that KIF13B is involved in the transport of secretory vesicles containing both PAUF and Rab6.

Analysis of vesicle fusion with the plasma membrane by total internal reflection fluorescence microscopy (TIRFM) revealed that KIF13B persists on the vesicles until the actual fusion event takes place and then disappears together with the Rab6 signal (*Figure 1G*). By performing fluorescence recovery after photobleaching (FRAP) experiments, we observed that GFP-KIF13B does not exchange on exocytotic vesicles (*Figure 1H*), similar to what we have previously observed for Rab6 and Rab8 (*Grigoriev et al., 2007*; *Grigoriev et al., 2011*). Our data indicate that the motor is stably bound to the vesicles, and its detachment is not required prior to vesicle fusion with the plasma membrane.

Next, we mapped the region of KIF13B responsible for the binding to Rab6 vesicles by testing the ability of KIF13B deletion mutants to associate with Rab6 vesicles in live cells. KIF13B contains an N-terminal motor domain (MD), a forkhead-associated (FHA) domain, several predicted coiled-coil regions (CC), a MAGUK binding stalk (MBS), two predicted domains of unknown function (DUF3694) and a C-terminal cytoskeleton-associated protein-glycine-rich (CAP-Gly) domain. The region

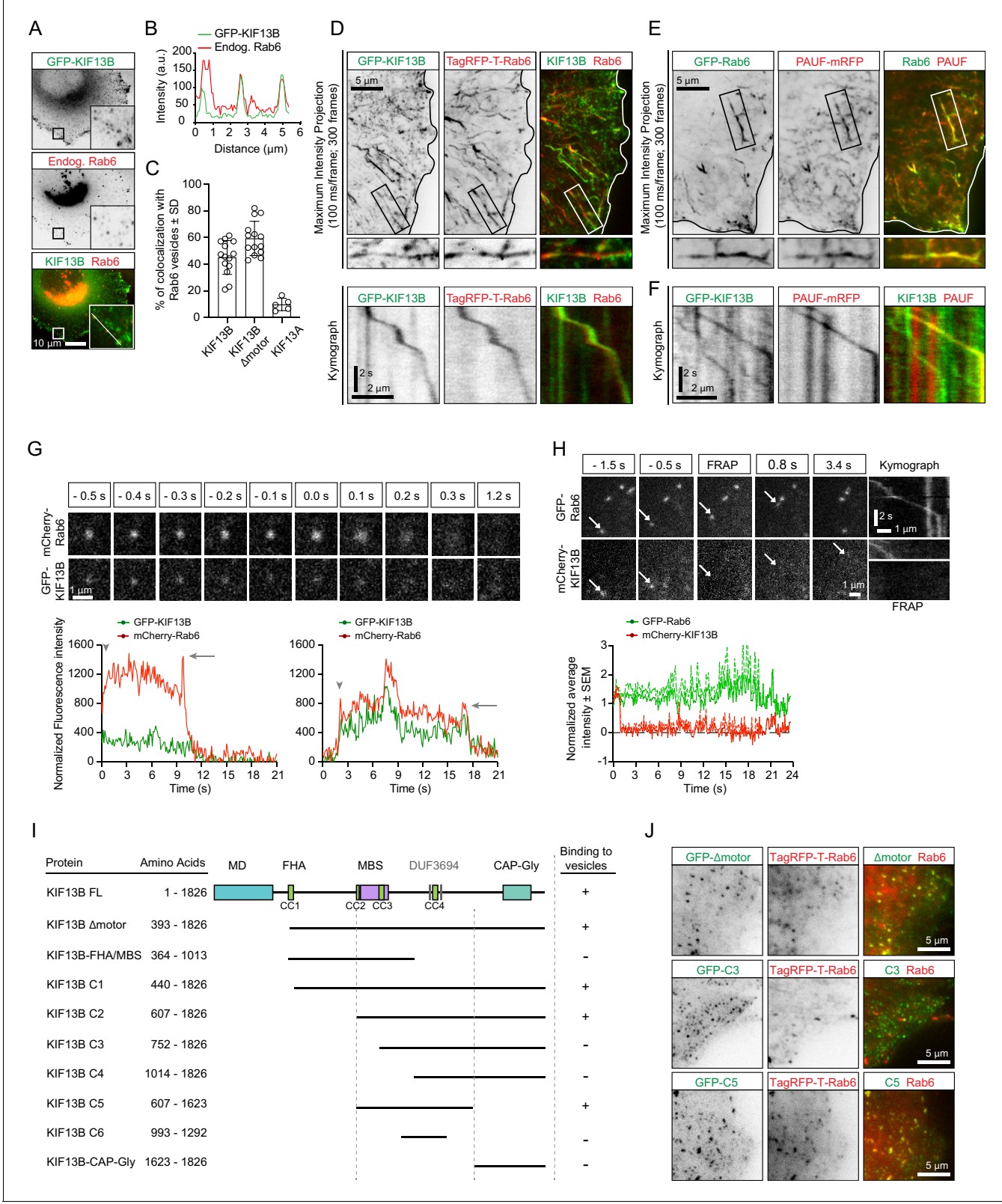

**Figure 1.** KIF13B localizes to Rab6-positive secretory carriers. (**A**) Immunofluorescence images of a HeLa cell expressing GFP-KIF13B and stained for endogenous Rab6. The insets correspond to magnified views of the boxed areas. (**B**) Signal intensity profile of GFP-KIF13B (green) and endogenous Rab6 (red) along the white line in panel A. (**C**) Quantification of the percentage of TagRFP-T labeled Rab6A vesicles colocalizing with GFP-KIF13B, GFP-KIF13B Δmotor and GFP-KIF13A. n = 14 (GFP-KIF13B, GFP-KIF13B Δmotor) and n = 5 cells (GFP-KIF13A). (**D**) Maximum intensity projections (300

*Figure 1 continued on next page*

*Figure 1 continued*

consecutive frames, 100 ms interval) illustrating imaging of live HeLa cells expressing GFP-KIF13B and TagRFP-T-Rab6A using TIRFM. Magnifications of the boxed area and kymographs illustrating the movement of co-labeled vesicle are shown below. (E) Maximum intensity projections with magnified views of the boxed areas illustrating TIRFM imaging of live HeLa cells transfected with GFP-Rab6A and PAUF-mRFP. (F) Kymographs illustrating the movement of vesicles labeled with GFP-KIF13B and PAUF-mRFP. (G) (Top) Frames from live TIRFM imaging showing the behavior of GFP-KIF13B and mcherry-Rab6A vesicles before and during fusion. 0.0 s corresponds to the sharp increase of fluorescent signal associated with vesicle docking at the plasma membrane. (Bottom) Two examples showing the average fluorescence intensity of a single vesicle labeled with GFP-KIF13B and mCherry-Rab6A plotted versus time. Vesicle appearance in the focal plane is indicated by an arrowhead. Arrow points to the peaks of fluorescence intensity prior to vesicle fusion with the plasma membrane. (H) (Top) TIRFM imaging combined with FRAP was performed in live HeLa cells containing exocytotic vesicles labeled for GFP-Rab6A and mCherry-KIF13B. The mCherry signal was photobleached (frames labeled FRAP) on moving vesicles labeled with GFP-Rab6A. Arrows indicate the same vesicle over time. Kymographs are shown to illustrate the absence of fluorescence recovery of mCherry-KIF13B on the vesicle. (Bottom) Quantification of the signal intensity of mCherry-KIF13B (red) on moving GFP-Rab6A vesicle (green) over time after FRAP. n = 6 vesicles in 4 cells. (I) Scheme of the GFP-KIF13B deletion constructs used in this study. The constructs were transfected in HeLa cells and colocalization with TagRFP-T-Rab6A-positive vesicles was determined by live cell imaging. The amino acid positions in KIF13B are indicated. MD, motor domain; FHA, forkhead-associated domain, MBS, MAGUK binding stalk; DUF, domain of unknown function; CC, coiled-coil. (J) Live images of HeLa cells expressing TagRFP-T-Rab6A and the indicated GFP-KIF13B deletion construct using TIRF microscopy.

The online version of this article includes the following source data and figure supplement(s) for figure 1:

**Source data 1.** An Excel sheet with numerical data on the fluorescence intensity profile of Rab6 and KIF13B on vesicles, colocalization of KIF13B, KIF13B Δmotor and KIF13A with Rab6 vesicles, fluorescence intensity profile of Rab6 and KIF13B on vesicles during fusion, and FRAP dynamics of Rab6 and KIF13B on vesicles represented as plots in *Figure 1B,C,G,H*.

**Figure supplement 1.** Analysis of localization of different markers to Rab6 vesicles.

**Figure supplement 1—source data 1.** An Excel sheet with numerical data on the quantification of the colocalization of Rab6 with PAUF-positive vesicles, and the fluorescence intensity profile of Rab6 and Eg5 on vesicles represented as plots in *Figure 1—figure supplement 1B,C*.

including the MBS, a coiled-coil and the DUF3694 domains (607–1623) was necessary and sufficient for the binding to Rab6 vesicles, whereas the CAP-Gly domain and the motor domain were dispensable for the binding (*Figure 1I,J*). Together, our results show that kinesin-3 KIF13B robustly and specifically interacts with PAUF/Rab6-positive exocytotic vesicles.

## KIF5B and KIF13B are the main drivers of Rab6 vesicle transport in HeLa cells

To critically test the role of different kinesins in the transport of secretory vesicles, we used CRISPR/Cas9 technology to knock out kinesin-encoding genes in HeLa cells. We generated gene knockout cell lines lacking KIF5B, KIF13B, KIF5B and KIF13B together (KIF5B/KIF13B-KO), or KIF5B, KIF13B, KIF1B and KIF1C (4X-KO), and observed that the silencing of these kinesin genes led to the loss of the respective proteins (*Figure 2A*). All obtained cell lines were viable and showed no proliferation defects. For kinesin-1 KIF5B knockout, these data are in line with previous genetic inactivation work in mice, which showed that although the mouse is embryonically lethal, extraembryonic cells are viable (*Tanaka et al., 1998*). Different kinesin knockout cells could spread normally as formation of focal adhesions was not perturbed, though the total cell area showed some variability between different cell lines, possibly due to clonal effects (*Figure 2—figure supplement 1A,B*). The MT cytoskeleton was somewhat less focused around the MT-organizing center in all cells lacking KIF5B, whereas the staining for EB1 comets appeared normal (*Figure 2—figure supplement 1A*). Endosomes showed increased perinuclear clustering, particularly in the 4X-KO cells, and their number appeared reduced in this cell line, although we cannot exclude that this effect was due to the fact that strongly clustered endosomes are more difficult to count (*Figure 2—figure supplement 1A,C*). Furthermore, mitochondria were shifted toward the perinuclear region in all cell lines lacking KIF5B (*Figure 2—figure supplement 1A,D*), as described previously (*Hooikaas et al., 2019*; *Tanaka et al., 1998*). Altogether, we generated a robust cellular system to study the individual contribution of several kinesin motors to intracellular transport.

To quantify the effects of kinesin knockouts on Rab6 vesicle motility, we performed automated vesicle tracking using an improved version of a particle tracking algorithm described previously (*Yao et al., 2017*; *Figure 2B*). Due to the very high Rab6 fluorescence signal, the Golgi area was manually excluded from analysis. Trajectories of individual Rab6 vesicles obtained through this analysis (termed here vesicle 'tracks') consisted of alternating periods of diffusive 'jiggling' movement and persistent directional motion, which we attributed to the active motor-driven transport along MTs.

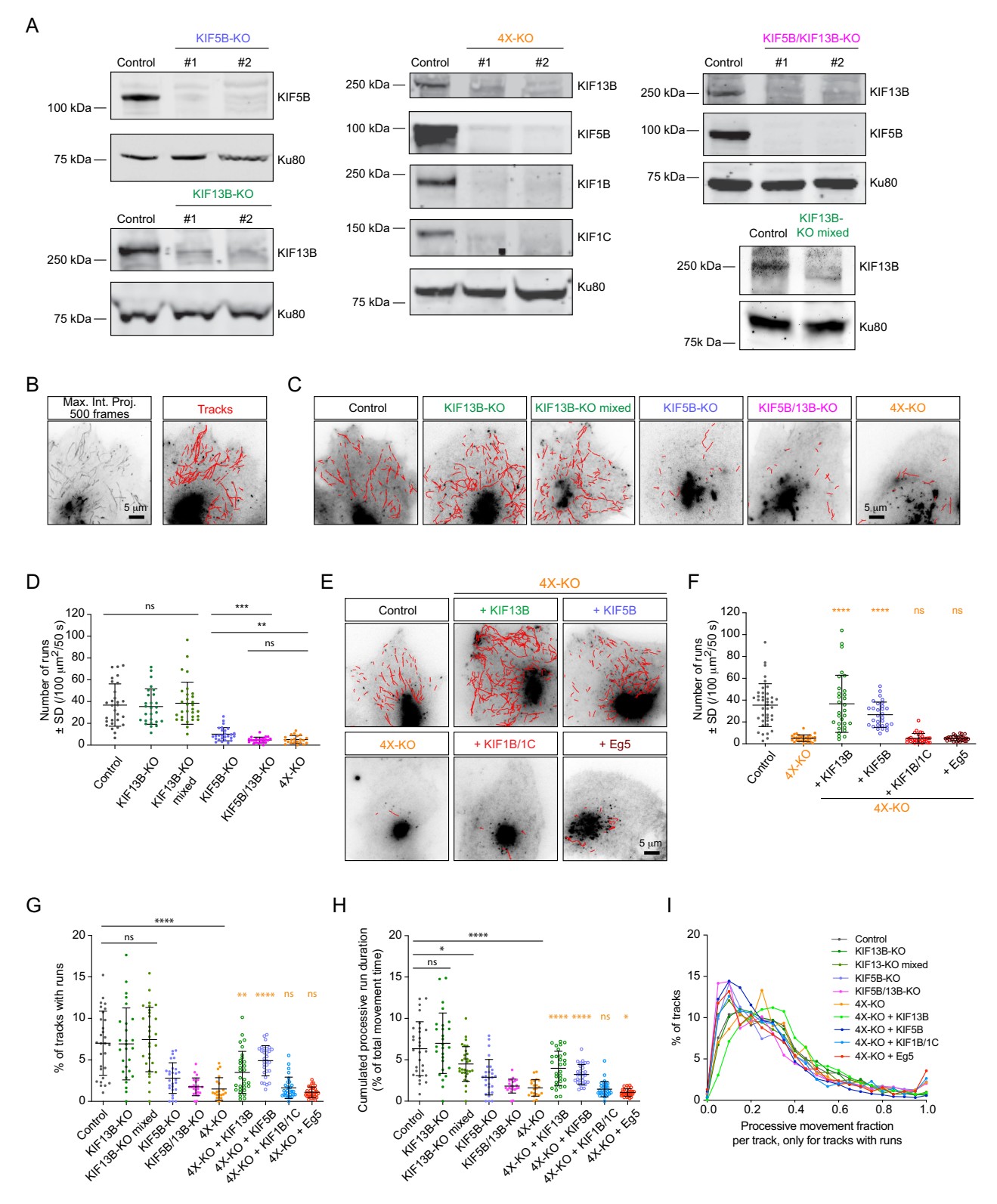

**Figure 2.** Transport of Rab6 vesicles in HeLa cells is driven by KIF5B and KIF13B. (**A**) Western blots of HeLa cell extracts showing the knockout of the different kinesins, using the indicated antibodies. Ku80 antibody was used as loading control. KIF13B-KO clone #1, KIF5B-KO clone #1, KIF5B/KIF13B-KO clone #1 and 4X-KO clone #2 have been used in latter experiments. (**B**) A maximum intensity projection over 500 frames (100 ms exposure with no delays) illustrating mCherry-Rab6A vesicle movement imaged by TIRFM and automated tracking using the SOS/MTrackJ plugin of mCherry-Rab6A

*Figure 2 continued on next page*

*Figure 2 continued*

labeled vesicles in the same HeLa cells. (C,E) Examples of automatic tracking of mCherry-Rab6A labeled vesicles using the SOS/MTrackJ plugin in the indicated knockout HeLa cells (C), or in 4X-KO cells expressing the indicated kinesins. (D) Analysis of the number of vesicle runs in the conditions depicted in (C). n = 29, 27, 30, 27, 22 and 21 cells, respectively, in three independent experiments. Mann-Whitney U test: \*\*\*p=0.0001; \*\*p=0.0015; ns, no significant difference. (F) Analysis of the number of vesicle runs in the conditions depicted in (E). n = 40, 30, 30, 30, 30 and 30 cells in three independent experiments for each condition except for Eg5 (two independent experiments). Mann-Whitney U test: \*\*\*\*p<0.0001; ns, no significant difference. (G) Tracks containing processive runs as a percentage of total number of Rab6 tracks per cell. n = 29, 27, 30, 27, 22, 21, 30, 30, 30 and 30 cells, respectively, in two independent experiments for Eg5 and 3 independent experiments in all other conditions. Mann-Whitney U test: \*\*\*\*p<0.0001; \*\*p=0.0011; ns, no significant difference. (H) Cumulated (total, summed) duration of processive runs as a percentage of total duration of all Rab6 tracks, per cell. The dataset is the same as in (G). Mann-Whitney U test: \*\*\*\*p<0.0001; \*p<0.047; ns, no significant difference. (I) Distribution of the fraction of processive motion to the total duration of a track. Only the tracks that contain processive runs are included. n = 4237, 3532, 5219, 1115, 494, 556, 5289, 3811, 833 and 842 tracks. The dataset is the same as in (G).

The online version of this article includes the following source data and figure supplement(s) for figure 2:

**Source data 1.** An Excel sheet with numerical data on the quantification of the effect of kinesin silencing and rescue on the number of vesicle runs, percentage of tracks with runs, cumulated processive run duration and the distribution of processive movements per track represented as plots in *Figure 2D,F–I*.

**Figure supplement 1.** Validation of kinesin knockout cell lines.

**Figure supplement 1—source data 1.** An Excel sheet with numerical data on the quantification of the effect of kinesin silencing on cell area, number of EEA1 vesicles and mitochondria clustering represented as plots in *Figure 2—figure supplement 1B–D*.

**Figure supplement 2.** Characterization of kinesin knockout cell lines.

**Figure supplement 2—source data 1.** An Excel sheet with numerical data on the quantification of Rab11 enrichment in the Golgi area in 4X-KO and in 4X-KO re-expressing KIF1B/1C, the effect of Eg5 depletion on the number of vesicle runs, and the effect of kinesin silencing and rescue on the average track duration with runs represented as plots in *Figure 2—figure supplement 2D,E,G*.

From complete vesicle tracks we extracted the segments of directional movement (here termed 'runs') using the velocity vector as a directional correlation measure, as described previously (*Katrukha et al., 2017*). Such processive runs represented less than 10% of the total vesicle motility; a detailed analysis of the share of processive movements in different conditions is discussed below. We compared the average number of vesicle runs in control and in the different knockout cell lines (*Figure 2C,D*; *Figure 2—figure supplement 2A*). In KIF13B-KO cells, the number of vesicle runs was very similar to control. Since some compensatory changes could have occurred during cell line selection of the clonal line, we also included in the analysis the population of cells that were used for KIF13B-KO clone selection (KIF13B-KO mixed population). These cells were transfected with a vector co-expressing a puromycin resistance gene, the single guide RNA sequence specific for KIF13B and Cas9, subjected to puromycin selection for 2 days and left to recover for 5 days before analysis (*Figure 2A*; 'KIF13B-KO mixed'). Also these cells did not show a major reduction in the number of runs, similar to KIF13B knockout clones (*Figure 2C,D*). In contrast, a very strong reduction in the number of runs was observed in the KIF5B-KO cells. Interestingly, the double knockout of KIF5B and KIF13B (KIF5B/KIF13B-KO) had a synergistic effect, as the number of Rab6 vesicle runs was reduced compared to the single KIF5B-KO (*Figure 2C,D*). These results indicate that both kinesins cooperate in the transport of Rab6 vesicles, with KIF5B being the major player. This is also in line with our previous observations showing that siRNA-mediated depletion of KIF5B had a strong impact on the post-Golgi transport of Rab6 vesicles (*Grigoriev et al., 2007*). The 4X-KO cells did not show a stronger reduction in the number of runs compared to the double KIF5B/KIF13B-KO (*Figure 2C,D*, *Figure 2—figure supplement 2A*), suggesting that KIF1B and KIF1C are not sufficient for Rab6 vesicle transport.

Next, we performed rescue experiments using the 4X-KO cells. We re-expressed in these cells KIF5B, KIF13B, a combination of KIF1B and KIF1C, or Eg5/KIF11, and analyzed Rab6 vesicle motility. Expression of KIF5B restored the number of vesicle runs to values similar to control, and the same was observed when KIF13B was re-expressed (*Figure 2E,F*). It has been previously shown that the knockdown of kinesin-1 heavy chain reduces the levels of kinesin light chains (KLC) (*Zhou et al., 2018*). Using immunofluorescence cell staining and Western blotting, we confirmed this observation (*Figure 2—figure supplement 2B*). KLC levels were restored by transiently re-expressing KIF5B-GFP in the KIF5B-KO line (*Figure 2—figure supplement 2C*). Interestingly, simultaneous re-expression of KIF1B and KIF1C did not rescue Rab6 vesicle motility in 4X-KO cells (*Figure 2E,F*). Importantly, these kinesins were functional, as their expression could rescue the localization of Rab11-positive

endosomes: compared to control cells, the distribution of Rab11 carriers was strongly shifted toward the Golgi area in the 4X-KO cells, and this phenotype was reversed upon simultaneous re-expression of KIF1B and KIF1C (*Figure 2—figure supplement 2D*). Furthermore, expression of Eg5 in the 4X-KO did not restore vesicle movement (*Figure 2E,F*), consistent with our observations that Eg5 does not localize to Rab6-positive vesicles (*Figure 1—figure supplement 1C,D*). In agreement with these data, the number of vesicle runs was not affected by siRNA-mediated depletion of Eg5 (*Figure 2—figure supplement 2E,F*). Thus, these data indicate that Eg5 and KIF1B/1C make no major contribution to the motility of Rab6 vesicles.

The data described above show that the number of processive Rab6 vesicle runs in HeLa cells depends on two motors, KIF5B and KIF13B. To get a more complete picture of vesicle motility in all the analyzed conditions, we investigated the share of processive runs compared to total Rab6 vesicle movements. We found that in all cases the percentage of tracks containing runs was below 10% (*Figure 2G*), and, as could be expected, it strongly correlated with the absolute number of vesicle runs (*Figure 2D,F*). A similar trend was observed when the total duration of processive motility was analyzed as a percentage of total vesicle movements (*Figure 2H*). This was due to the fact that the fraction of processive movement inside the tracks that did contain runs remained almost the same for all conditions (on average, one third; *Figure 2I*). In all conditions, the duration of tracks that contained runs was consistently longer (on average, six times) than tracks without runs (*Figure 2—figure supplement 2G*). Together, these data indicate that KIF5B and KIF13B drive a large fraction of processive movements of Rab6 vesicles but have no major impact on their diffusive motility. It should be noted, however, that since our analysis is optimized for extracting processive vesicle runs, additional work will be needed for the precise characterization of diffusive vesicle movements, as it is highly sensitive to tracking errors occurring during particle linking between frames.

## KIF5B and KIF13B have different velocities and set the speed range of Rab6 vesicles

Next, we used automated vesicle tracking to examine how the presence of KIF5B and KIF13B affects different parameters of vesicle movement. Vesicle speeds obtained from the automated analysis were very similar to those determined manually from the analysis of kymographs drawn along Rab6-positive tracks, as we described previously (*Grigoriev et al., 2007*; *Schlager et al., 2014*; *Figure 3—figure supplement 1A*). Taking this approach, we examined the speed of the residual vesicle movements in different knockout cell lines. While the average Rab6 vesicle speed in KIF13B-KO cell clones and the mixed KIF13B-KO population was mildly but significantly reduced compared to control, the few Rab6 vesicles that still moved in the KIF5B-KO, KIF5B/KIF13B-KO and 4X-KO cells displayed speeds much higher than those in control cells, an effect that was particularly obvious when speeds of all individual tracks rather than average velocities per cell were plotted (*Figure 3A,B*; *Figure 2—figure supplement 2A*). Interestingly, we observed the appearance of vesicles moving with speeds exceeding 3 µm/s, which were very rare in control cells but could be readily detected in KIF5B-KO, KIF5B/KIF13B-KO and 4X-KO cells (*Figure 3B*, *Figure 3—figure supplement 1B*). The residual, faster movements of Rab6 vesicles in cells lacking KIF5B and KIF13B could be driven by another kinesin or by dynein, as our previous work has shown that dynein-dependent Rab6 vesicle movements can occur with velocities of 2–4 µm/s (*Schlager et al., 2014*; *Splinter et al., 2012*). We also analyzed other parameters of Rab6 vesicle motility, namely the duration and length of individual vesicle runs. We found that run duration was relatively constant in all conditions (*Figure 3—figure supplement 1B*). Loss of KIF13B had no significant impact on run length, indicating that this motor does not significantly contribute to the processivity of vesicle movement driven by KIF5B. Interestingly, the residual runs observed in the absence of KIF5B were significantly longer than those in control cells (*Figure 3—figure supplement 1B*), possibly reflecting the strong sensitivity of this motor to obstacles (*Telley et al., 2009*).

We next analyzed the distribution of Rab6 vesicle speeds in more detail, focusing on the conditions where the number of vesicle movements was reasonably high (4X-KO cells and their rescues with KIF1B/C and Eg5 were not included in this analysis because the overall number of Rab6 vesicle movements in these cells was too low to fit speed distributions in a reliable manner). If only one motor were involved in vesicle movement, one Gaussian curve would be expected to fit the data. However, in most cases, the distributions of vesicle speeds matched much better the sum of two Gaussians (*Figure 3—figure supplement 1C*, *Figure 3C,D*), indicating that there are at least two

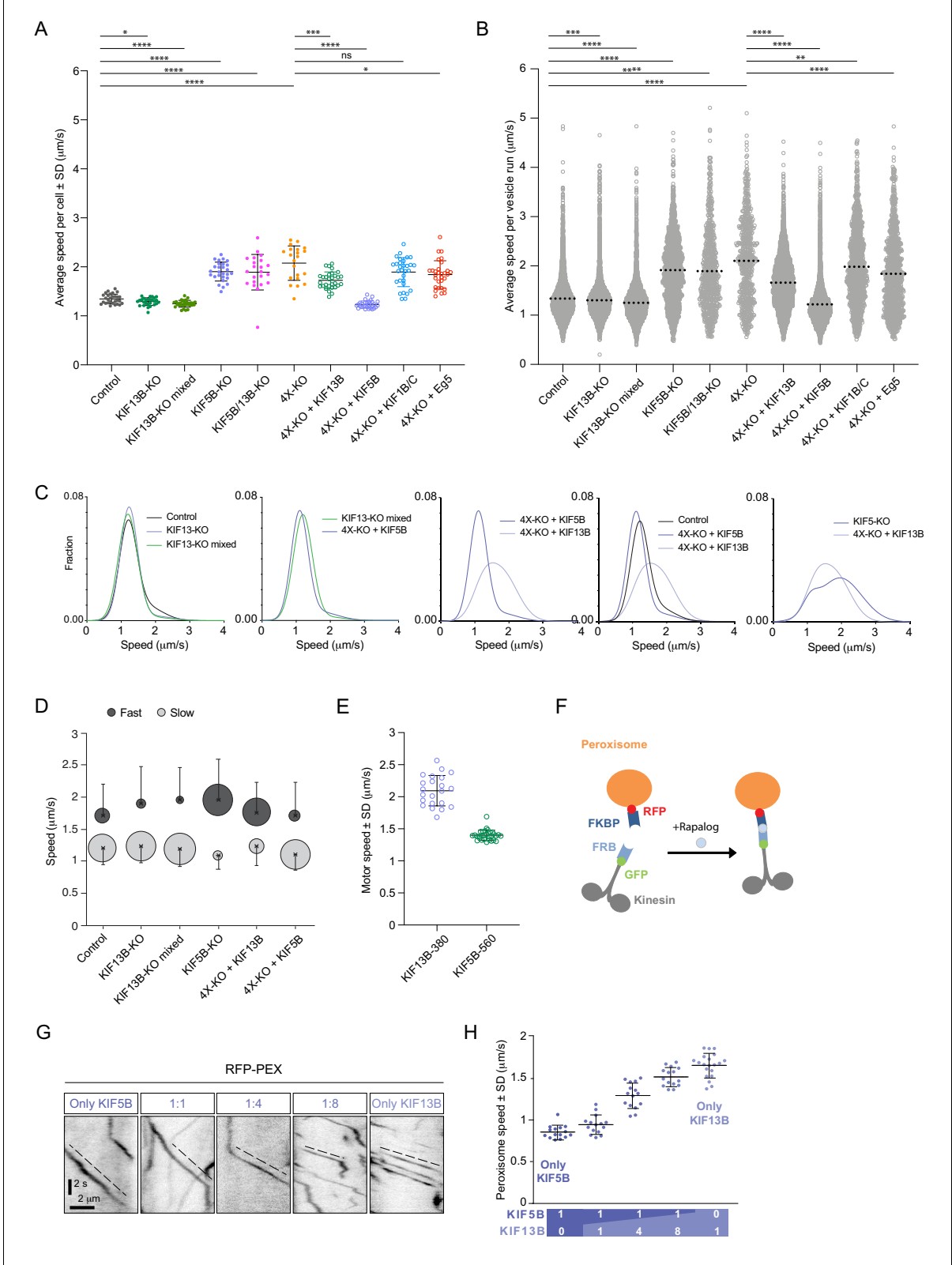

**Figure 3.** KIF5B and KIF13B have distinct speeds. (**A**) Analysis of the mean speed of automatically tracked mCherry-Rab6A-positive vesicles per cell in the indicated conditions. n = 29, 27, 30, 27, 22, 21, 30, 30, 30 and 30 cells, respectively, in two independent experiments for Eg5 and 3 independent experiments in all other conditions. Unpaired t-test: ****p<0.0001; ***p=0.0002; *p<0.025, ns, no significant difference. (**B**) Violin plots showing the speed of individual automatically tracked mCherry-Rab6A-positive vesicles in the indicated conditions. Dotted lines represent the mean. n = 29, 27, 30,

*Figure 3 continued on next page*

*Figure 3 continued*

27, 22, 21, 30, 30, 30 and 30 cells, respectively, in two independent experiments for Eg5 and 3 independent experiments in all other conditions. Same data as shown in (A), but displayed here for individual vesicle runs rather than averaged per cell. Mann-Whitney U test: ****p<0.0001; ***p=0.0002; **p=0.0064. (C) Combinations of sums of two Gaussian fits to the distribution of Rab6 vesicles run speeds for the indicated conditions (see *Figure 3—figure supplement 1C*). (D) Parameters of 'slow' (light gray) and 'fast' (dark gray) Gaussians components from fits in *Figure 3—figure supplement 1C*. Crosses correspond to the mean value and error bars to the standard deviation of fitted Gaussians. The area of the circles corresponds to the fraction of runs associated with 'slow' and 'fast' components, i.e. represents the area under the curve of each Gaussian (total sum of areas for each condition is the same). n = 7099 runs in 29 cells, 6682 runs in 27 cells, 8192 runs in 30 cells, 1772 runs in 27 cells, 7616 runs in 30 cells and 5651 runs in 30 cells, respectively. (E) Speed of kinesin-positive particles imaged with TIRF microscopy in 4X-KO HeLa cells expressing KIF5B-560-GFP (n = 30 cells, three experiments) or KIF13B-380-LZ-GFP (n = 22 cells, two experiments). (F) A scheme of inducible peroxisome trafficking assay performed by the rapalog-dependent recruitment of FRB-tagged KIF5B(1-807) or KIF13B(1-444) to FKBP-tagged PEX3, a peroxisome protein. (G) Kymographs illustrating peroxisome movements in cells transfected with the indicated FRB-tagged KIF5B(1-807):KIF13B(1-444) plasmid ratios. (H) MRC5 cells were co-transfected with PEX3-mRFP-FKBP and the indicated ratios of FRB-tagged KIF5B and KIF13B plasmids mentioned in (F) simultaneously, while the total amount of DNA was kept constant. Peroxisomes were imaged by TIRF or SD microscopy and their speeds were measured 10–40 min after rapalog addition. n = 15, 15, 15, 15 and 20 cells, respectively in three experiments (Kif13B alone - in four experiments).

The online version of this article includes the following source data and figure supplement(s) for figure 3:

**Source data 1.** An Excel sheet with numerical data on the quantification of the effect of kinesin silencing and rescue on vesicle speed, KIF13B-380 and KIF5B-650 motor speed, and peroxisome speed in cells expressing different ratios of KIF5B and KIF13B represented as plots in *Figure 3A–E,H*.
**Figure supplement 1.** Characterization of Rab6 vesicle motility in kinesin knockout lines.
**Figure supplement 1—source data 1.** An Excel sheet with numerical data on the quantification of Rab6 vesicle speed distribution from manual and automated analysis, and the effect of kinesin silencing on the distribution of speed, duration and length of vesicle runs represented as plots in *Figure 3—figure supplement 1A–C*.

populations of vesicles with different average speeds per run. In control, there is a slower vesicle population with speed 1.20 ± 0.26 µm/sec (average ± SD), and a faster one with a value of 1.71 ± 0.49 µm/s (average ± SD) (*Figure 3—figure supplement 1C*, *Figure 3C,D*). The fraction of higher speed runs was reduced in the clonal and mixed KIF13B-KO cells, while it was increased in the KIF5B-KO and KIF13B rescue of 4X-KO (*Figure 3C,D*, *Figure 3—figure supplement 1C*). The speed distributions were quite similar in KIF13B-KO cells and in 4X-KO cells rescued with KIF5B (*Figure 3C*, second plot), in line with idea that KIF5B is the major motor on both cases.

In contrast, Rab6 vesicles in 4X-KO cells rescued by expressing KIF13B moved significantly faster than those in cells rescued with KIF5B (*Figure 3C,D*, *Figure 3—figure supplement 1C*). It is important to emphasize that in cells rescued by expressing either KIF5B or KIF13B, vesicle tracks were distributed throughout the whole cytoplasm (*Figure 2E*). Thus, the observed speed difference could not be explained by differences in cell regions where the movement took place, such as cell thickness or interference with other organelles. However, the observed difference in vesicle speeds can be explained by previous observations showing that KIF13B is a faster motor compared to KIF5B (*Arpağ et al., 2014*; *Norris et al., 2014*). Indeed, single-molecule imaging of dimeric motor-containing fragments lacking the cargo binding tails, KIF13B-380-LZ and KIF5B-560, in 4X-KO cells confirmed that KIF13B moves significantly faster than KIF5B (*Figure 3E*). To show that this difference is also observed when these kinesins are linked to cargo, we next used the FRB-FKBP chemical dimerization system in combination with rapalog (an analog of rapamycin) to trigger the binding of KIF5B (1-807)-GFP-FRB and/or KIF13B(1-444)-GFP-FRB motor-containing fragments to peroxisomes, which are relatively immobile organelles (*Kapitein et al., 2010b*; *Splinter et al., 2012*; *Figure 3F–H*). We used different ratios of KIF5B and KIF13B-expressing plasmids as a way to manipulate the relative motor abundance on peroxisomes and observed that peroxisome speed increased when the KIF5B:KIF13B plasmid ratio decreased (*Figure 3G,H*), similar to what we described previously for KIF5B and KIF1A (*Gumy et al., 2017*). Kymographs of peroxisome movements showed that when the two kinesins were co-expressed, intermediate velocities were observed, suggesting simultaneous engagement or very rapid switching between the two types of motors (*Figure 3G*). This behavior is different from that previously described for the kinesin-1/kinesin-3 motor pairs connected by a stiff linker, where typically only one of the two kinesins was engaged at a given moment and switching between motors could be observed (*Norris et al., 2014*). Interestingly, a significant excess of KIF13B-encoding plasmid was needed to shift peroxisome speeds to higher values (*Figure 3G,H*), in agreement with in vitro gliding assays, which indicated that the slower kinesin-1 predominated when combined with kinesin-3 at a 1:1 molar ratio (*Arpağ et al., 2014*).

Next, we imaged fluorescent fusions of the full-length KIF5B and KIF13B motors in 4X-KO cells. The expression of GFP-tagged kinesins in combination with mCherry-Rab6A in 4X-KO cells allowed us to directly determine the speeds of vesicles driven by a particular motor and also to estimate the number of kinesins present on the vesicles in our rescue experiments (*Figure 4A–D*). By manually measuring the speeds of mCherry-Rab6A labeled vesicles decorated with either GFP-KIF13B or KIF5B-GFP in 4X-KO cells, we again confirmed that KIF13B is the faster motor (*Figure 4C,D*). To determine the number of motors present on a vesicle, we used KIF5B-560-GFP, a dimeric KIF5B fragment that lacks the tail and therefore does not bind to cargo, as a fluorescence intensity standard (*Figure 4A*). We observed that approximately 1–2 dimers of KIF5B-GFP and 3–5 dimers of GFP-KIF13B could be detected on the vesicles (*Figure 4A,B*). Since these experiments were performed with overexpressed full-length kinesins, these data suggest that the maximum number of KIF5B binding sites on Rab6 vesicle is lower than that of KIF13B. However, it is likely that in the endogenous situation the KIF13B abundance is low and the KIF13B binding sites are not saturated. This would explain why overexpressed but not endogenous KIF13B can rescue Rab6 vesicle motility in the absence of KIF5B. This would also explain why Rab6 vesicle speed distributions in KIF5B-KO cells and 4X-KO cells rescued with KIF13B differ: in 4X-KO cells rescued with KIF13B this motor predominates, while in KIF5B-KO cells, the population of vesicles moving with the speed characteristic of KIF13B is relatively small, whereas the other residual movements in these cells are driven by even faster motors such as dynein (*Figure 3C,D*, *Figure 3—figure supplement 1C*).

Finally, we co-expressed KIF5B-GFP and mCherry-KIF13B together and found that they could be present simultaneously on the same Rab6-positive vesicle (*Figure 4E,F*). We confirmed that the colocalization of the three fluorescent signals was not due to bleed-through between the channels (*Figure 4—figure supplement 1*). We developed an algorithm to automatically detect movements of two colocalized markers (see next section and Materials and Methods) and used it to measure vesicle speeds. In 4X-KO cells where the two kinesins were expressed separately, this analysis showed that mCherry-Rab6A vesicles bound to KIF5B-GFP moved slower than the ones bound to GFP-KIF13B, whereas in co-expressing cells, the particles associated with both motors had speeds that were intermediate between those of KIF5B and KIF13B (*Figure 4F,G*), in agreement with the data on the recruitment of these two motors to peroxisomes (*Figure 3H*).

## KIF5B and KIF13B are differently distributed on moving Rab6 vesicles

We next set out to test whether the distribution of the motors on moving cargo correlates with the movement direction and therefore can be used to infer motor activity. We reasoned that the motors linked to the lipid bilayer will redistribute to the front of the vesicle if they are pulling it, will localize at the rear of the vesicle if they generate a hindering force or will display no shift if they are not engaged. We co-expressed GFP-KIF13B or KIF5B-GFP together with mCherry-Rab6A in 4X-KO HeLa cells and imaged vesicle motility using TIRFM (*Figure 5A,B*). PAUF-mRFP, in combination with GFP-Rab6A, was used as a control of localization of a vesicle marker lacking motor activity. The positions of the centers of two fluorescent signals on each vesicle were determined with sub-pixel localization precision using 2D Gaussian fitting. The alignment of the two fluorescent channels and the sub-pixel correction of chromatic aberrations were performed using a calibration photomask (*Figure 5—figure supplement 1A*). Vesicle and motor trajectories (*Figure 5C*) were separated into phases of directed and random movement (*Figure 5D,E*), and only the periods of colocalized directional runs were used for further analysis. To describe the relative positions of the fluorescent markers during movement, we determined the projection of the vector $\vec{l}$ between the two signals on the axis defined by vesicle's velocity vector and termed it projected distance $d$ (*Figure 5F*). This value is positive if a marker is at the front of a moving vesicle and negative if it is at the back. As expected, the projected distances between Rab6 and PAUF were symmetrically distributed around zero (*Figure 5G*). In contrast, the projected distances between KIF5B or KIF13B and Rab6 were strongly skewed toward positive values, with a median value of 76 and 56 nm, respectively (*Figure 5G*). The angles between the line connecting the centers of the two fluorescent signals (the distance vector) and the velocity vector (angle $\alpha$, *Figure 5F*) were distributed randomly when the positions of Rab6 and PAUF were analyzed. In contrast, in the case of Rab6 and the two kinesins, the angles close to zero predominated, as can be expected if the kinesins were accumulating at the front of the moving vesicles (*Figure 5H*).

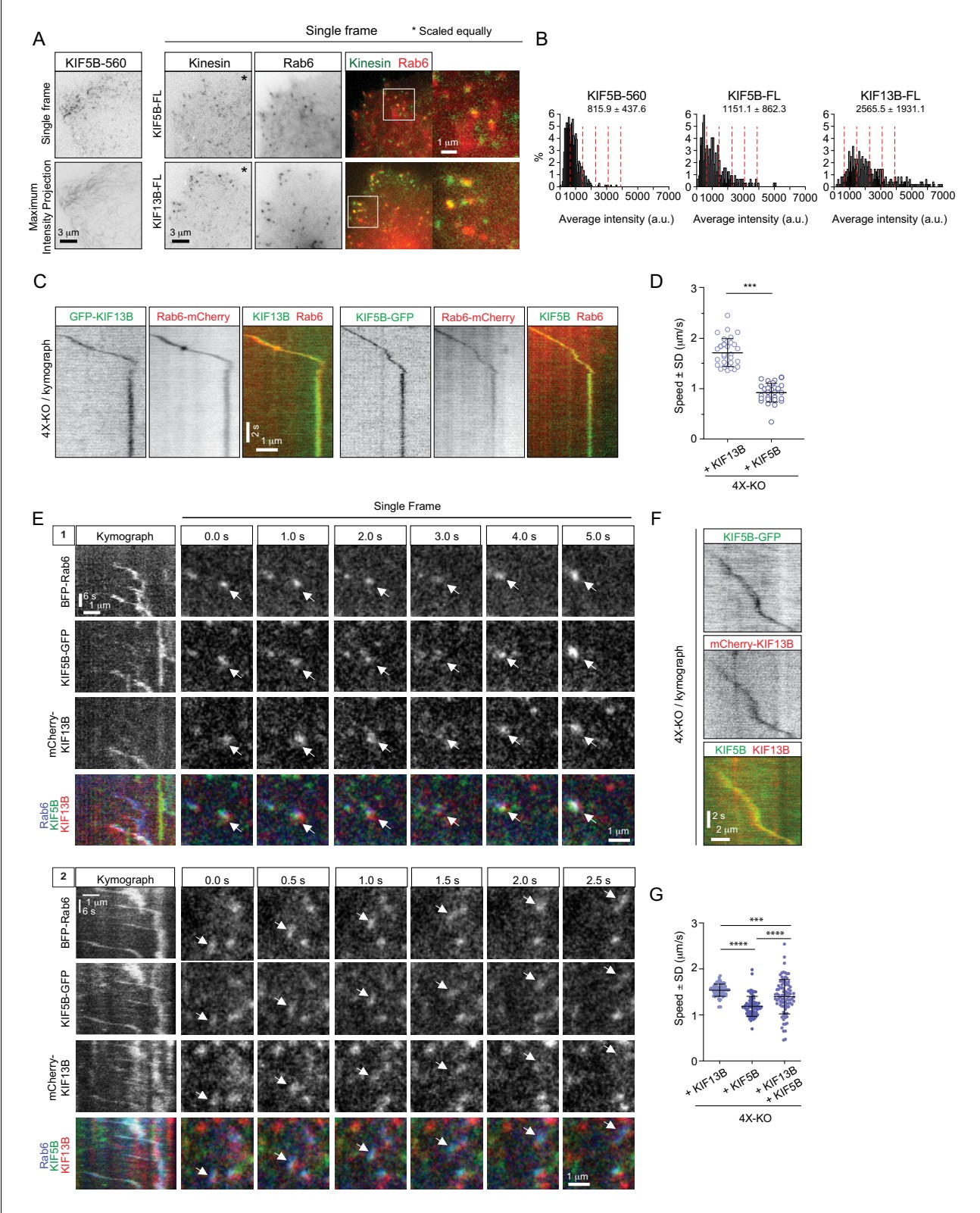

**Figure 4.** KIF5B and KIF13B colocalize on moving vesicles. (**A**) Live TIRFM imaging of 4X-KO cells expressing either KIF5B-560-GFP alone or mCherry-Rab6A together with the full-length (FL) KIF5B-GFP or GFP-KIF13B. A single frame and a maximum intensity projection of a KIF5B-560-GFP movie is shown on the left, and single frames illustrating the localization of KIF5B-FL-GFP and GFP-KIF13B-FL on mCherry-Rab6A-labeled vesicles are shown on the right. Insets show enlargement of boxed areas. (**B**) Histograms showing the frequency distributions of the average intensity of the indicated kinesin

*Figure 4 continued on next page*

*Figure 4 continued*

constructs in 4X-KO cells annotated with the mean ± SD; for KIF5B-FL-GFP and GFP-KIF13B-FL, only the signals colocalizing with Rab6 vesicles were quantified. n = 660 in 17 cells, 320 in 24 cells and 495 in 19 cells in two independent experiments (KIF5B-560) and three independent experiments (KIF5B-FL and KIF13B-FL). Dashed lines mark intensity of 1, 2, 3, 4 and 5 kinesin molecules estimated from the average value of KIF5B-560 distribution. (C–D) 4X-KO HeLa cells expressing mCherry-Rab6A together with GFP-KIF13B-FL or KIF5B-FL-GFP were imaged using TIRFM. Kymographs illustrating the movement of Rab6 vesicles positive for KIF13B or KIF5B were drawn and used to manually measure the velocity of each motor. n = 30 and 28 cells in three independent experiments. Mann-Whitney U test: ***p<0.001. (E) Live confocal imaging of 4X-KO cells co-expressing TagBFP-Rab6, full-length KIF5B-GFP and full-length mCherry-KIF13B. Kymographs of two different moving vesicles and corresponding consecutive single frames (1 and 2) are shown to illustrate the simultaneous localization of KIF5B-GFP and mCherry-KIF13B on moving TagBFP-Rab6-positive vesicles. (F) A kymograph illustrating the movement of a particle co-labeled with KIF5B-GFP and mCherry-KIF13B and expressed in a 4X-KO HeLa cell. (G) Automated analysis of speeds of Rab6 vesicles colocalized with indicated full-length kinesins in 4X-KO HeLa cells expressing either KIF5B-GFP and mCherry-KIF13B alone, or the condition where two kinesins colocalize together. Dots show average speed per cell, bars represent mean and SD. n = 82, 79 and 89 cells, same data and analysis as in *Figure 5G, H*. Mann-Whitney U test: ****p<0.0001; ***p=0.0006.

The online version of this article includes the following source data and figure supplement(s) for figure 4:

**Source data 1.** An Excel sheet with numerical data on the quantification, in 4X-KO cells, of the distribution of the average intensity of fluorescently tagged KIF5B-560, KIF5B-FL and KIF13B-FL, the speed of KIF13B and KIF5B, and the speed of vesicles containing KIF13B, KIF5B or both motors represented as plots in *Figure 4B,D,G*.

**Figure supplement 1.** Validation of simultaneous three-color image acquisition.

Analysis of individual vesicles showed that the maximal projected distance between Rab6 and KIF5B or KIF13B signals observed for a given vesicle varied from a few tens of nanometers to ~500 nm (*Figure 5I*), as expected given that vesicles can have different sizes. Since the average projected distances between KIF13B and Rab6 signals were smaller than those between KIF5B and Rab6 signals (*Figure 5G*), it appears that smaller vesicles are generated when the 4X-KO cells are rescued with KIF13B compared to KIF5B. This observation was confirmed when we measured absolute distances between Rab6 and other markers without projecting them on the velocity direction (*Figure 5J*).Importantly, vesicles detected by imaging PAUF and Rab6, without overexpressing any motors, were in the same size range as those found in 4X-KO cells rescued with KIF5B, whereas vesicles in 4X-KO cells rescued with KIF13B were smaller (*Figure 5J*). In addition, more vesicles with larger areas were observed when 4X-KO cells were rescued with KIF5B compared to KIF13B (*Figure 5K*). These data suggest that KIF5B, which is normally the main motor transporting Rab6 vesicles, is important for controlling vesicle size, a function that cannot be compensated by KIF13B, possibly because it performs less well under hindering load (*Arpağ et al., 2019*; *Arpağ et al., 2014*). We could not detect any dependence of the distance between Rab6 and the kinesins on vesicle speed (*Figure 5I*; *Figure 5—figure supplement 1B*).

Next, we compared the distribution of KIF5B-GFP and mCherry-KIF13B when they were colocalized on the same moving particle (*Figure 5G,H*, green line). Interestingly, in this situation, KIF5B was shifted to the rear compared to KIF13B, which would be consistent with the idea that KIF5B, which is slower and has a lower detachment probability under load (*Arpağ et al., 2019*; *Arpağ et al., 2014*), exerts a drag force, whereas the faster KIF13B accumulates at the front of the vesicle. This is consistent with the fact that the average speeds of particles bearing both motors are slower than those transported by kinesin KIF13B alone, but faster than those moved by KIF5B (*Figure 4G*).

As the large majority of Rab6 vesicles are transported toward cell periphery and thus MT plus ends, we assumed that most of the displacements we analyzed represent kinesin-driven runs. However, there is a small fraction of Rab6 vesicles that are transported to the MT minus-ends by dynein (*Grigoriev et al., 2007*; *Matanis et al., 2002*) and, therefore, we next analyzed in more detail the behavior of kinesins on vesicles moving in the minus-end direction. To identify such runs, we automatically selected trajectories where a vesicle underwent a clear and acute switch of direction, and split such tracks into pairs of runs (for example, run1 and run2 or run2 and run3) where average movement direction was opposite (*Figure 5L* and Methods). We then calculated the projected distance *d* between the kinesin and Rab6 signals for each directional run within the run pair. The run with the higher projected distance *d* was assigned as the forward run, and the other run as the backward run (*Figure 5L,M*). The projected distance between kinesins and Rab6 in the forward runs was positive, as expected, and similar to that observed in the full dataset shown in *Figure 5G*, where the tracks were not selected for the presence of opposite polarity runs. Interestingly, for the backward

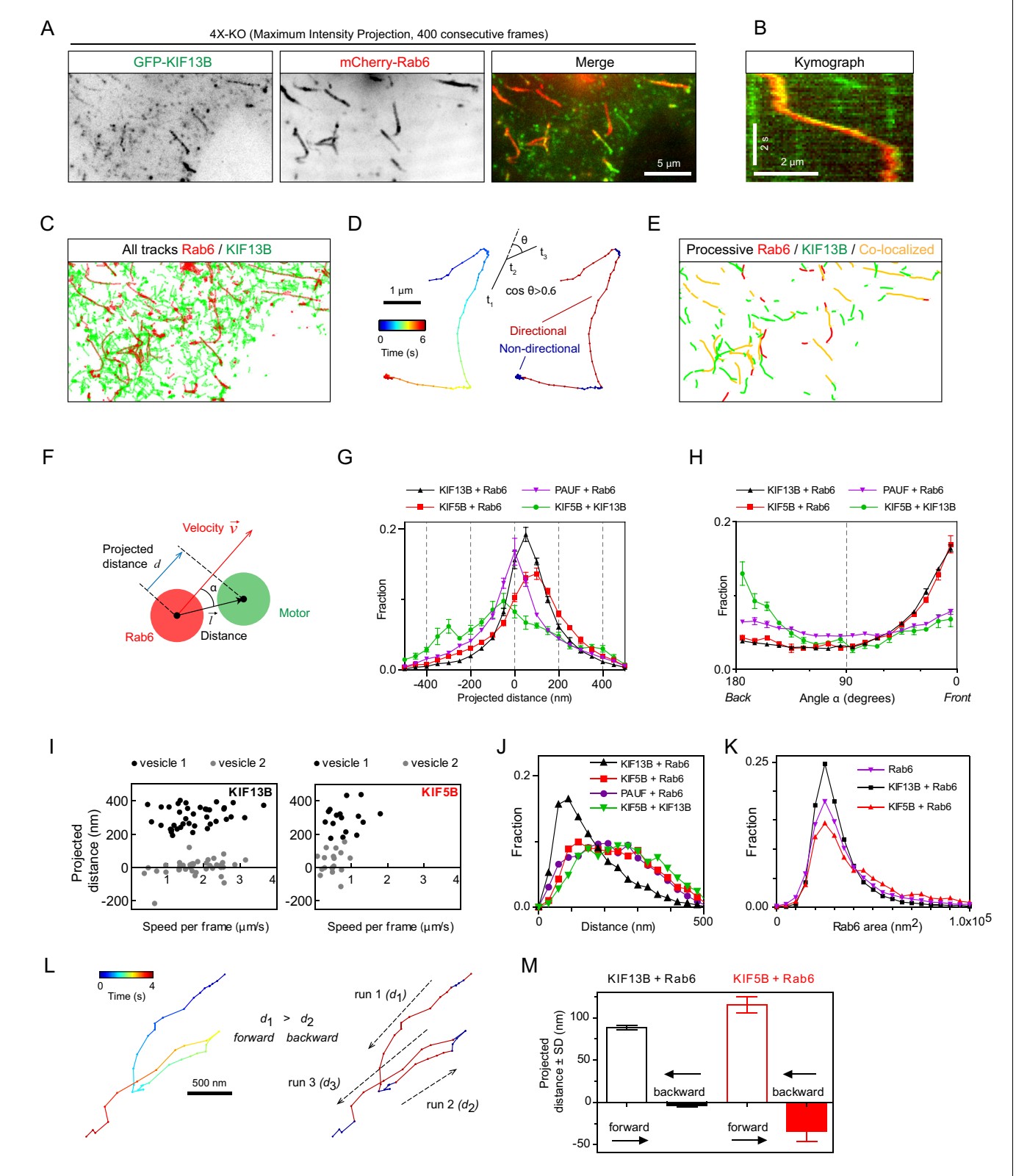

**Figure 5.** Kinesins exhibit distinct distributions on Rab6 vesicles. (**A**) A representative example of maximum intensity projection (400 consecutive frames, 100 ms interval) of 4X-KO HeLa cells expressing mCherry-Rab6A and GFP-KIF13B to visualize events of Rab6-vesicle movement. Chromatic aberration of the red channel (mCherry-Rab6A) was corrected based on calibration, as illustrated in *Figure 5—figure supplement 1A*. (**B**) A kymograph from the movie shown in (**A**) illustrating the movement of a mCherry-Rab6A-labeled vesicle positive for GFP-KIF13B. (**C**) Automatically extracted trajectories of

*Figure 5 continued on next page*

*Figure 5 continued*

mCherry-Rab6A- and GFP-KIF13B-positive particles (detected independently) from the movie shown in (A). (D) Segmentation of trajectories into periods of random and directed motion. (Left) An example of Rab6 vesicle trajectory with color-coded time. (Middle) Definition of directional movements: Movement was classified as directional when a cosine of the angle θ between two consecutive velocity vectors ($t_3$–$t_2$ and $t_2$–$t_1$) was larger than 0.6. (Right) Final segmentation result with directional (red) and random (blue) periods of movement. (E) Directional segments of the tracks shown in (C), with colocalizing tracks labeled in yellow. (F) Schematics of the parameters used to characterize the distribution of two markers on the same vesicle. The projected distance $d$ is calculated as a projection of distance between the centers of motor and cargo fluorescent signals ($\vec{l}$) onto the axis defined by the instant velocity vector ($\vec{v}$) of the cargo. The angle α is defined as the angle between the distance and velocity vectors. (G,H) The averaged histograms of the instantaneous projected distance (G) and the angle α (H) for GFP-KIF13B (black), KIF5B-GFP (red) with respect to mCherry-Rab6A, PAUF-mRFP with respect to GFP-Rab6A (purple) and for KIF5B-GFP with respect to mCherry-KIF13B (green). Each dot and bar represent the average and SEM over several independent experiments, each including 8–20 cells. KIF13B (N = 6 independent experiments, 82 cells, 11333 runs, 55129 time points), KIF5B (N = 6, 79 cells, 2826 runs, 10023 time points), KIF5B and KIF13B (N = 7, 89 cells, 1558 runs, 4371 time points) and PAUF (N = 2, 20 cells, 5807 runs, 21359 time points). (I) Plots of projected distance between Rab6A and KIF13B (left) or KIF5B (right) signals against speed for four different vesicles/runs (two different vesicles with distinct maximum projected distances, likely reflecting different vesicle sizes, are shown for each kinesin). (J,K) Histograms of the distance between the indicated markers (J) and Rab6A area (K) averaged per run and pooled together for all experiments. Same statistics as in (G,H). (L) Extraction of opposite polarity runs from Rab6 vesicle trajectories. On the left, an example of a trajectory with color-coded time; on the right, the same trajectory where the color denotes movement characteristics, directed (red) or random (blue). For each processive segment (run) the average direction of the velocity vector (dashed arrows) and average projected distance value ($d_i$) are calculated. Within one trajectory, the algorithm searches for all possible pair combination and keeps only those where the average movement direction is opposite. Within each pair a run with the higher average projected displacement is assigned to be the 'forward' run and the other one the 'backward' run. (M) Instantaneous (per frame) projected displacements for pairs of opposite runs, average ± SEM for the denoted conditions. The data are the same as in (G–J); for KIF13B, 664 opposite run pairs found, 3262 forward and 3122 backward time points; for KIF5B, 83 run pairs, 289 forward and 274 backward time points.

The online version of this article includes the following source data and figure supplement(s) for figure 5:

**Source data 1.** An Excel sheet with numerical data on the quantification of the distribution and projected displacement of KIF13B and KIF5B signals on moving vesicles represented as plots in *Figure 5G–K,M*.
**Figure supplement 1.** Analysis of kinesin distribution on Rab6 vesicles.
**Figure supplement 1—source data 1.** An Excel sheet with numerical data on the quantification of the projected displacement and angle relative to speed of KIF13B and KIF5B signals plotted as heat maps in *Figure 5—figure supplement 1B*.

---

runs, the average displacement was close to zero for KIF13B (~3.5 nm) and was slightly negative (~ -34.1 nm) for KIF5B (*Figure 5M*). These data suggest that when a vesicle switches to dynein-driven motility, KIF13B does not undergo any tug-of-war with dynein, whereas KIF5B might be exerting some hindering force. These data would be consistent with the observations that a kinesin-3 is readily detached under force while kinesin-1 is more resistant to force (*Arpağ et al., 2019*; *Arpağ et al., 2014*). Overall, our sub-pixel localization analysis suggests that both kinesins can actively engage on the vesicle when it moves toward MT plus end, but only KIF5B can oppose dynein.

## Kinesin-driven Rab6 vesicle transport spatially regulates secretion in mammalian cells

We next analyzed the functional importance of kinesin-mediated transport for secretion in HeLa cells. Previous work showed that partial depletion of KIF5B did not have a major effect on the overall secretion of neuropeptide Y (NPY), a soluble cargo present in Rab6 vesicles (*Grigoriev et al., 2007*). Now that we have established a system (the 4X-KO line) where Rab6 vesicle motility was dramatically suppressed, we reassessed the impact of kinesin-based transport of post-Golgi vesicles on the secretion levels. We took advantage of the retention using selective hooks (RUSH) system (*Boncompain et al., 2012*) to synchronize secretion. The interaction of SBP-GFP-E-Cadherin with streptavidin-KDEL (Hook) allows for the retention of E-Cadherin-GFP at the ER and its subsequent release for transport toward to the Golgi apparatus and the plasma membrane upon biotin addition (*Figure 6A*). We performed the RUSH assay by overexpressing SBP-GFP-E-Cadherin and streptavidin-KDEL in control or 4X-KO HeLa cells, and induced secretion by adding biotin. We performed live imaging and observed similar E-Cadherin accumulation at the plasma membrane in both conditions (*Figure 6B*). To quantify this, we specifically labeled surface-exposed E-Cadherin using a primary GFP/secondary Alexa641-conjugated antibody pair and used flow cytometry to analyze the intensity of the surface staining of E-Cadherin (Alexa641) against its total expression (GFP) (*Figure 6—figure supplement 1*). One hour after the addition of biotin, the surface staining of GFP in 4X-KO cells was

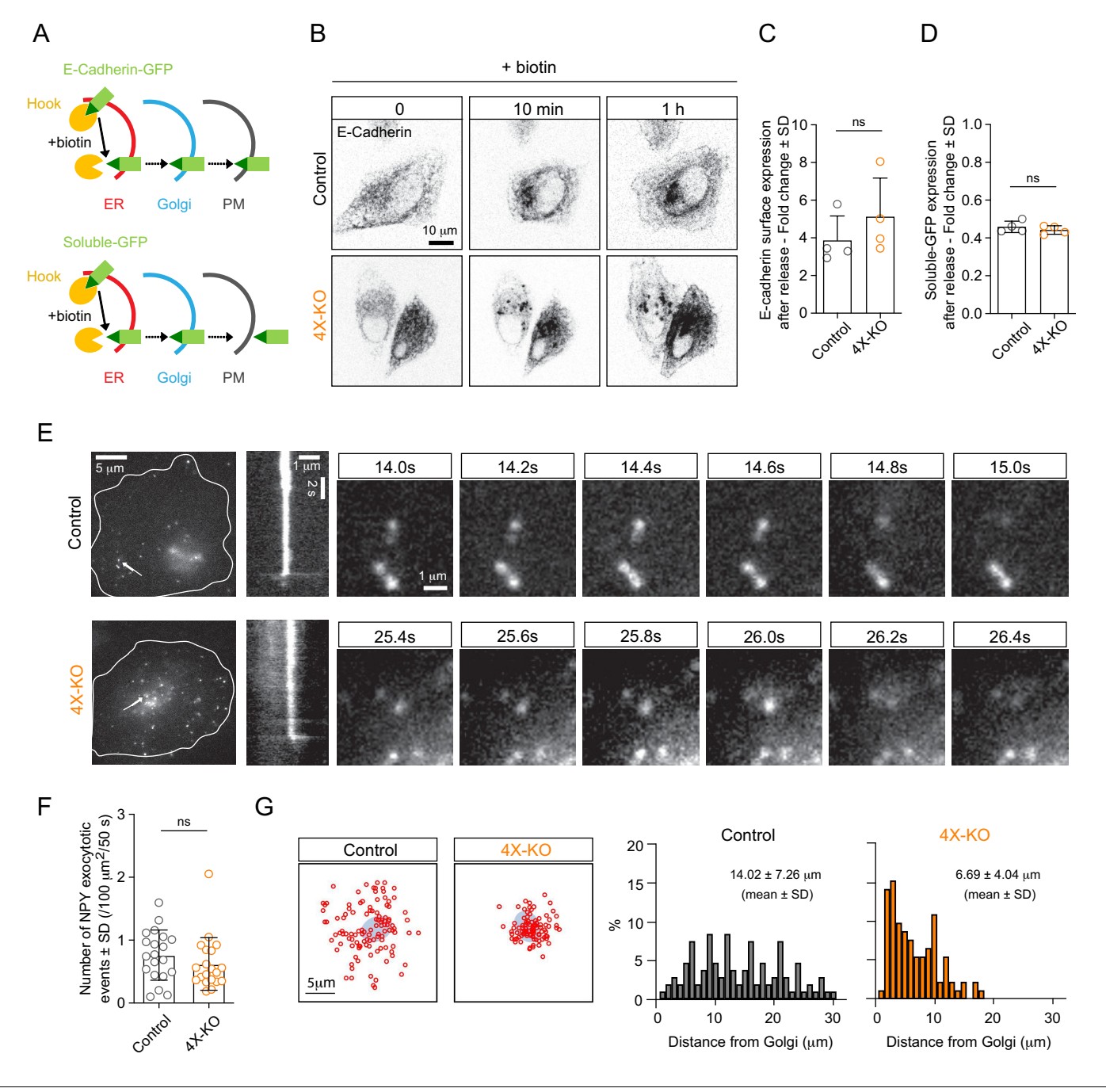

**Figure 6.** Kinesins control the spatial distribution but not the efficiency of exocytosis. (**A**) A scheme depicting the RUSH assay used in this study. The interaction of SBP-GFP-E-Cadherin or soluble-GFP-SBP with streptavidin-KDEL (Hook) allows for the retention of E-Cadherin-GFP or soluble-GFP in the ER and their release for transport to the Golgi and the plasma membrane (PM) upon the addition of biotin, which competes with SBP for streptavidin binding. (**B,C**) RUSH assay was performed by expressing SBP-GFP-E-Cadherin and streptavidin-KDEL from the same bicistronic expression plasmid in control or 4X-KO HeLa cells. Cells were treated with biotin and imaged using time-lapse spinning-disk confocal microscopy (B, GFP-E-Cadherin signal) or subjected to surface staining with anti-GFP antibody to specifically label plasma membrane-exposed E-Cadherin followed by flow cytometry analysis (C). Plot shows the fold change of Alexa641 mean intensity (surface staining) in E-Cadherin expressing cells before and after biotin addition (1 hr). n = 4 independent experiments. Mann-Whitney U test: ns, no significant difference. (**D**) Control or 4X-KO HeLa cells expressing soluble-GFP-SBP and streptavidin-KDEL were treated with biotin and analyzed by flow cytometry to quantify the fold change of GFP mean intensity after biotin addition. n = 4 independent experiments. Mann-Whitney U test: ns, no significant difference. (**E,F**) TIRF microscopy was used to visualize and analyze exocytosis events in control or 4X-KO HeLa cells expressing NPY-GFP. Exocytotic events, defined by a fast burst of fluorescence followed by the disappearance of the signal were visually identified, confirmed by kymograph analysis and counted per cell and per surface area and per duration of movie (50 s) (**F**)

*Figure 6 continued on next page*

*Figure 6 continued*

n = 20 cells in two independent experiments. Mann-Whitney U test: ns, no significant difference. Individual time frames in (E) illustrate representative exocytotic events; their localization is indicated by white arrows, and the corresponding kymographs are shown. (G) Analysis of the spatial distribution of the NPY exocytotic events shown in (E). Schematized positions of NPY exocytosis events (red circles) compared to the position of the Golgi (blue) are shown on the left (sum of 20 cells) and frequency distributions of the distance between the center of the Golgi and the sites of exocytosis in control and 4X-KO HeLa cells are shown on the right. n = 109 and 93 tracks from 20 cells in two independent experiments.

The online version of this article includes the following source data and figure supplement(s) for figure 6:

**Source data 1.** An Excel sheet with numerical data on the quantification of the effect of kinesin silencing on E-cadherin surface expression and soluble-GFP expression, the number of NPY exocytotic events and the distribution of the distance between the center of the Golgi and the sites of NPY exocytotic events represented as plots in *Figure 6C,D,F,G*.

**Figure supplement 1.** Quantification of secretion using flow cytometry.

similar to that in control cells (*Figure 6C*, *Figure 6—figure supplement 1*). To confirm these results, we used a different version of the RUSH system and analyzed the synchronized secretion of soluble-GFP upon biotin addition (*Figure 6A*). Similarly, control or 4X-KO HeLa cells expressing soluble-GFP-SBP and streptavidin-KDEL were treated with biotin and analyzed by flow cytometry. We did not observe any difference in the secretion of soluble-GFP two hours after biotin addition (*Figure 6D*).

We also used TIRFM to investigate the spatial distribution of exocytosis sites using NPY-GFP, a soluble Rab6 vesicle cargo. Events of NPY-GFP exocytosis are characterized by a burst of fluorescence followed by rapid signal disappearance (*Grigoriev et al., 2007*; *Figure 6E*). We did not observe significant differences in the number of exocytotic events in the 4X-KO compared to control cells (*Figure 6F*). However, whereas in control cells these events were distributed along the entire radius of the cell, with many events at the cell periphery, in 4X-KO cells exocytotic events were restricted to the vicinity of the Golgi (*Figure 6G*). Kinesin-driven Rab6-vesicle transport is thus not essential for secretion efficiency, but it is required for the correct spatial distribution of exocytotic events.

## Combination of KIF5B and KIF13B allows Rab6 vesicles to reach growing MT plus ends

Next, we set out to investigate the spatial regulation of the activity of the kinesins associated with Rab6 vesicles. Previous work has shown that KIF5B depends on the members of MAP7 family for its activation (*Hooikaas et al., 2019*; *Metzger et al., 2012*; *Monroy et al., 2018*; *Pan et al., 2019*), while kinesin-3 is inhibited by MAP7 (*Monroy et al., 2018*; *Monroy et al., 2020*). We used a previously described MAP7-KO HeLa cell line either alone or in combination with siRNAs against MAP7D1 and MAP7D3 to inhibit the expression of all three MAP7 homologues expressed in HeLa cells (*Hooikaas et al., 2019*; *Figure 7A*). The number of Rab6 vesicle runs was not significantly affected in MAP7-KO cells, but it was significantly reduced in cells lacking all MAP7 proteins (*Figure 7B,C*). These results are similar to those observed in KIF5B-KO cells (*Figure 2C,D*). The speed of Rab6 vesicle movement was similar in control and MAP7-KO cells, but increased in cells lacking all MAP7 proteins (*Figure 7D*), consistent with the results obtained in KIF5B-KO cells (*Figure 3A–D*; *Figure 3—figure supplement 1B,C*).

Simultaneous labeling of MTs with antibodies against α-tubulin and MAP7, MAP7D1 or MAP7D3 showed that the MAP7/tubulin intensity ratio decreases along the cell radius (*Figure 8A*), and this distribution was not affected by the knockout of KIF5B (*Figure 8—figure supplement 1*), indicating that transport driven by this kinesin has no detectable effect on the localization of these MAPs. We reasoned that MAP7 enrichment may contribute to the activation of KIF5B in the central part of the cell, and, given the fact that KIF5B is a slower motor compared to KIF13B, this would result in slower velocity of Rab6 vesicle transport close to the Golgi compared to the cell periphery. Indeed, the increase of Rab6 vesicle velocity observed in cells lacking MAP7 proteins was more pronounced in the center of the cell compared to the cell periphery (*Figure 8B*). Furthermore, the absence of KIF13B, the faster motor, led to a decrease in Rab6 vesicle speed at the cell margin (*Figure 8B*), where the contribution of this motor would be expected to be more substantial. Conversely, overexpression of KIF13B in the 4X-KO cell line led to an increase in Rab6 vesicle speed along the cell radius.

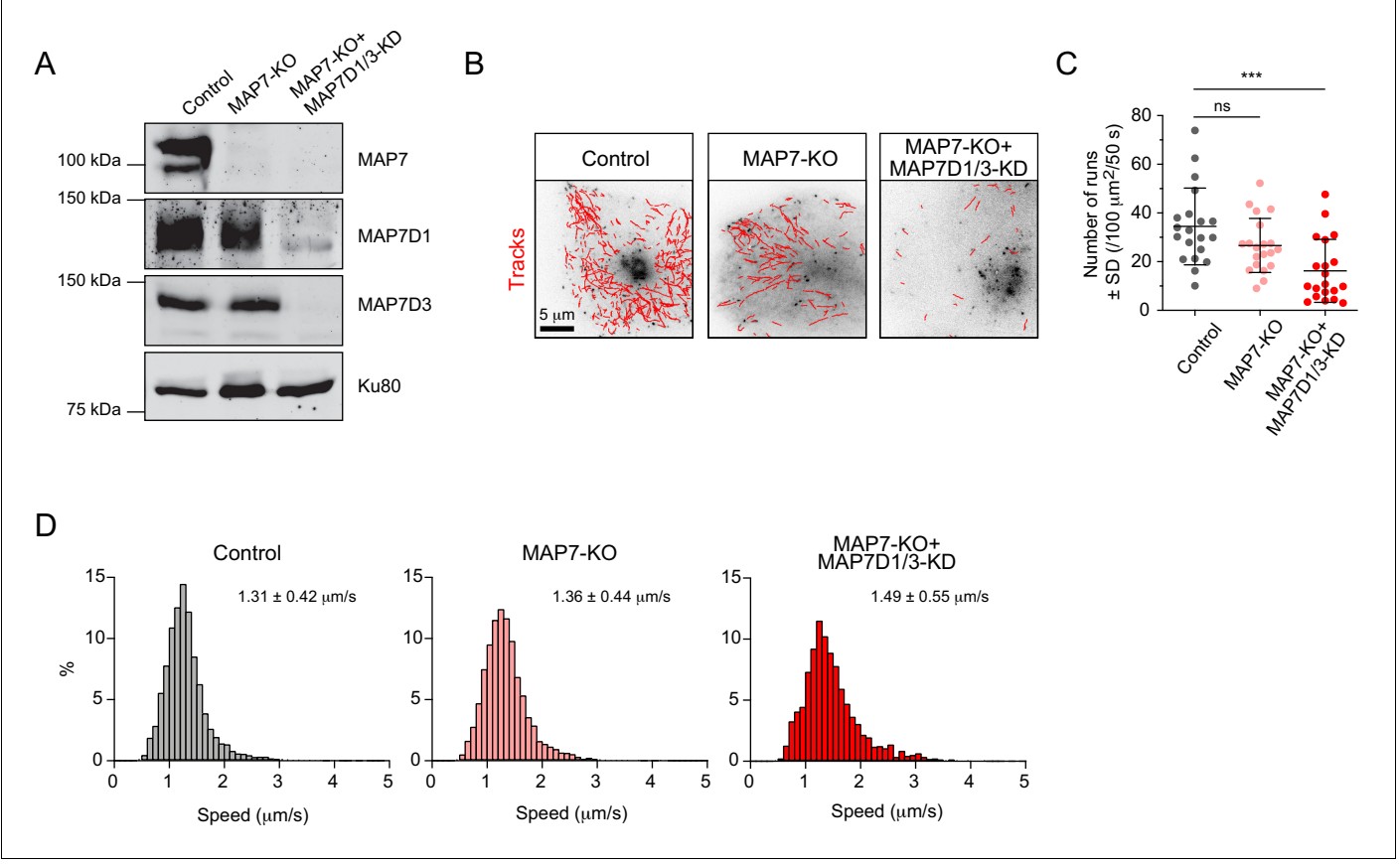

**Figure 7.** MAP7 family proteins are required for the transport of Rab6 vesicles. (**A**) Western blot analysis of the extracts of control or MAP7 knockout (MAP7-KO) HeLa cells or MAP7-KO cells transfected with siRNAs against MAPD1 and MAP7D3 (MAP7-KO+MAP7D1/3-KO) with the indicated antibodies. (**B–D**) GFP-Rab6A was expressed and imaged using TIRFM in Hela cells described in (**A**). Automatic tracking using the SOS/MTrackJ plugin (500 consecutive frames, 100 ms interval) of the Rab6A signal (**B**), number of Rab6 vesicle runs per cell, n = 20 cells in two experiment in each condition (**C**) and the frequency distributions of Rab6 vesicle speeds after automatic tracking annotated with the mean ± SD (**D**) are shown. n = 5038, 4056 and 2366 tracks from 20 cells in two independent experiments. Mann-Whitney U test: ***p=0.0002, ns, no significant difference.

The online version of this article includes the following source data for figure 7:

**Source data 1.** An Excel sheet with numerical data on the quantification of the effect of MAP7 family proteins silencing on the number of Rab6 vesicle runs and on the distribution of Rab6 vesicle speeds represented as plots in *Figure 7C,D*.

The gradient of MAP7 protein distribution along the cell radius could be caused by MT dynamics: freshly polymerized MT segments are enriched at the cell periphery, and they can be expected to be less populated by MAP7 proteins if MAP7 binding to MTs is relatively slow. Indeed, live cell imaging showed that MAP7 displays a delay in binding to freshly grown, EB3-positive MT ends (*Figure 8C,D*). Also KIF5B followed this localization pattern: the dimeric motor fragment KIF5B-560 was significantly less abundant on EB3-positive MT ends compared to older, EB3-negative MT lattices (*Figure 8E,F*). In contrast, the motor domain containing fragment of KIF13B, KIF13B-380, which was dimerized by fusing it to a leucine zipper, was enriched at the growing MT ends (*Figure 8G,H*). Based on these results, we hypothesized that in the absence of KIF13B, Rab6 vesicles will be less efficient in reaching the newly grown, EB3 positive MT ends and found that this was indeed the case (*Figure 8I,J*). This effect was obvious in the mixed KIF13B-KO but was less apparent in the clonal KIF13B-KO cell lines, possibly because compensatory changes may have occurred during clone selection. Altogether, our results suggest that the combined action of KIF5B and KIF13B allows Rab6 vesicles to efficiently navigate different MT populations. MAP7 proteins contribute to the spatial regulation of exocytotic vesicle transport by promoting the activation of KIF5B and possibly by inhibiting KIF13B on older MTs.

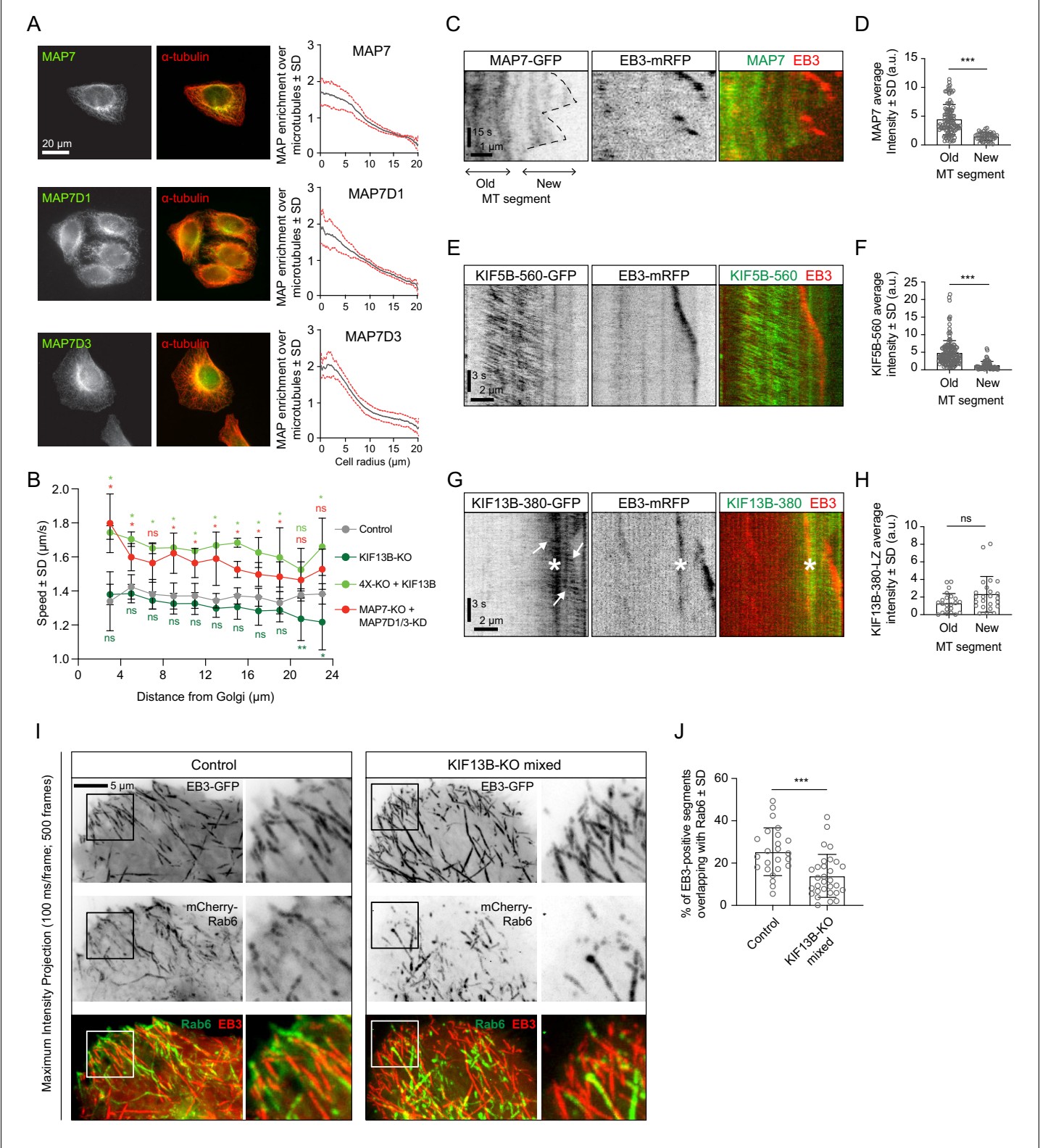

**Figure 8.** KIF13B promotes Rab6 vesicle transport to freshly polymerized MT ends. (**A**) Staining of HeLa cells for endogenous MAP7, MAP7D1 or MAP7D3 together with α-tubulin and quantification of the relative enrichment of MAP7 protein signal intensity over the tubulin signal calculated and plotted against the distance from the cell center. n = 50, 48 and 41 cells, respectively, from 4, 4 and 3 independent experiments. (**B**) Automatic tracking of GFP-Rab6A or mCherry-Rab6A labeled vesicles was performed on data obtained by TIRFM imaging in control, KIF13B-KO, KIF13B-expressing 4X-KO

*Figure 8 continued*

or MAP7-KO+MAP7D1/3-KD HeLa cells. Tracking results were analyzed using the SAID plugin of MTrackJ as described in the Methods to extract the velocity of the Rab6 runs in relation to their distance from the Golgi. n = 49, 47, 30 and 40 cells, respectively, in 5, 5, 3 and 4 independent experiments. Mann-Whitney U test: **p=0.0079, *p<0.032, ns, no significant difference. (C, E, G) Kymographs illustrating the dynamics of a MT labeled with EB3-mRFP together with MAP7-GFP (C), KIF5B-560-GFP (E) or KIF13B-380-LZ-GFP (G) imaged using TIRFM. In (G), arrows indicate movement of KIF13B-380-LZ-GFP along MTs and the asterisk (*) marks the spot where another EB3-mRFP/KIF13-380-LZ-GFP-positive MT crosses the analyzed MT. (D, F, H) Quantification of the average intensity of MAP7-GFP (D), KIF5B-560-GFP (F) and KIF13B-380-LZ-GFP (H) on old (further than 3 μm from plus end) and new (within 2 μm from plus end) MT segments visualized with EB3-mRFP (mean ± SD). D: n = 109 (14 cells in two independent experiments) and n = 47 (14 cells in two experiments). F: n = 155 (19 cells in two independent experiments) and n = 71 (19 cells in two experiments). H: n = 24 (15 cells in three experiment) and n = 24 (15 cells in three experiment). Mann-Whitney U test: ***p<0.0001, ns, no significant difference. (I) Maximum intensity projections (500 consecutive frames, 100 ms interval) of control or KIF13B-KO mixed HeLa cells expressing mCherry-Rab6A and EB3-GFP. Magnified views of the boxed areas are shown on the right. Colors in the merged images were inverted for display purposes. (J) Quantification of the percentage of EB3-GFP-positive MT segments colocalizing with mCherry-Rab6A vesicles. Measuring has been performed in MetaMorph software using Measure Colocalization option on Threshold images. n = 24 and 30 cells, respectively, in three independent experiments. Mann-Whitney U test: ***p<0.0003.

The online version of this article includes the following source data and figure supplement(s) for figure 8:

**Source data 1.** An Excel sheet with numerical data on the quantification of MAP7 family proteins enrichment on MTs along the cell radius, the effect of kinesin and MAP7 family proteins silencing on vesicle speed relative to distance from the Golgi, MAP7, KIF5B-560 and KIF13B-380 intensity on old and new MT segments, and the effect of KIF13B silencing on the co-localization of EB3-GFP-positive MT segments with Rab6 vesicles represented as plots in *Figure 8A,B,D,F,H,J*.

**Figure supplement 1.** KIF5B knockout does not affect the localization of MAP7 family proteins.

**Figure supplement 1—source data 1.** An Excel sheet with numerical data on the quantification of the effect of KIF5B silencing on MAP7 family proteins enrichment on MTs along the cell radius represented as plots in *Figure 8—figure supplement 1*.

## Discussion

In this study, we have dissected kinesin-driven transport of Rab6-positive secretory carriers and the behavior of kinesins on these carriers. We showed that in HeLa cells, kinesin-1 KIF5B and kinesin-3 KIF13B are the main motors driving secretory vesicle motility. Rab6 vesicle transport was profoundly perturbed in a knockout cell line lacking KIF5B, KIF13B, KIF1B and KIF1C (4X-KO) and could be restored by re-expression of KIF5B and KIF13B but not KIF1B or KIF1C. KIF5B is the predominant motor driving Rab6 vesicle motility in HeLa cells: its knockout impairs vesicle movement much stronger than that of KIF13B, and the average speed of Rab6 vesicle movement in control cells is more similar to that of KIF5B than of KIF13B. This is likely due to the relatively low abundance of KIF13B in HeLa cells, because when overexpressed, KIF13B alone can restore Rab6 vesicle motility, in line with the previous studies showing that KIF13B and its *Drosophila* counterpart, Kinesin-73, are super-processive kinesins that can drive long-distance cargo transport (*Huckaba et al., 2011*; *Siddiqui and Straube, 2017*; *Soppina et al., 2014*). Efficient rescue of Rab6 vesicle movement with KIF13B in 4X-KO cells suggests that defective motility of Rab6 vesicles is due to the lack of motors attached to these vesicles rather than indirect effects caused by the lack of KIF5B and its impact on MTs or other organelles.

Rab6 represents a robust marker of a broad variety of post-Golgi carriers (*Fourriere et al., 2019*; *Grigoriev et al., 2007*). Such vesicles are likely to be heterogeneous, and the composition of motors controlling their motility is expected to be cell type-specific. Indeed, our previous work established the involvement of KIF1A/B/C family in the transport of Rab6-positive carriers in neurons and Vero cells (*Gumy et al., 2017*; *Schlager et al., 2010*). Kinesin recruitment to the vesicles depends on the expression of specific adaptor proteins. KIF5B was recently shown to be bound to different membranes including Rab6 vesicles by the Dopey1-Mon2 complex, which functions as a lipid-regulated cargo-adaptor (*Mahajan et al., 2019*). KIF1C was shown to interact with Rab6-positive membranes directly (*Lee et al., 2015*) and also by associating with a cell-specific adaptor protein BICDR-1, which is absent from HeLa cells (*Schlager et al., 2010*). How KIF13B is recruited to Rab6 vesicles is currently unclear, but once recruited, it stays stably attached to the vesicle. Interestingly, our experiments with overexpression of fluorescently tagged kinesins in 4X-KO cells indicate that the number of KIF5B binding sites on a Rab6 vesicle is more limited than that for KIF13B, suggesting that the two kinesins use different adaptors. It is possible that the adaptor complexes for KIF5B and KIF13B on Rab6 vesicles have some overlapping components, similar to lysosomes, where the BORC/Arl8 complex recruits KIF5B through the interaction with SKIP and KLC and binds to kinesin-3 KIF1A/B

directly, through the interaction with the coiled-coil domains of the motors (*Guardia et al., 2016*). However, spatial segregation of KIF5B and KIF13B present on the same vesicles (see below) suggests that they are not part of the same molecular complex.

Performing rescue experiments in kinesin knockout cells allowed us to generate cells where all kinesins driving Rab6 vesicle transport were fluorescently labeled, thus enabling us to study their distribution on cargo. We found that when expressed separately in 4X-KO cells, both KIF5B and KIF13B localized at the front of moving vesicles; however, when the two motors were co-expressed, KIF5B tended to be positioned behind KIF13B on moving particles. This can be explained by the distinct speeds of kinesin-1 and kinesin-3, described in previous studies (*Arpağ et al., 2014*; *Guedes-Dias et al., 2019*; *Soppina et al., 2014*) and confirmed here: KIF13B, being a faster motor, relocates to the front of the vesicle, while KIF5B, being slower, stays behind. The higher resistance of KIF5B to detachment under load, which is thought to be the main determinant of motor behavior in a mixture of motors (*Arpağ et al., 2014*), also likely contributes to this distribution. The reported interaction between KIF5B and the dynein adaptor BICD2 (*Grigoriev et al., 2007*) or some additional interactions between KIF5B and the dynein-dynactin complex could in principle control the localization of this kinesin on the vesicle. Importantly, we find that KIF5B tends to move to the rear of the vesicle when it switches directions and is presumably driven to MT minus ends by cytoplasmic dynein. These data are incompatible with the idea that KIF5B and dynein are part of the same protein complexes on secretory vesicles. Our data further suggest that KIF5B, which is attached to the vesicles independently of dynein, is engaged when dynein is driving transport toward the minus ends, and this could also be explained by KIF5B's resistance to hindering load. Thus, unlike KIF13B, which detaches under hindering load, KIF5B can engage in a tug-of-war with the dynein motor.

The generation of 4X-KO cells also allowed us to critically test the functional importance of kinesin-mediated transport for constitutive secretion. In line with previous work showing that drug-mediated MT disassembly does not block post-Golgi trafficking (*Fourriere et al., 2016*; *Hirschberg et al., 1998*; *Rindler et al., 1987*; *Rogalski et al., 1984*; *Van De Moortele et al., 1993*), we found that the efficiency of secretion was normal is cells lacking four kinesins. Previous work suggested that kinesin-1 contributes to the generation of Rab6 vesicles by pulling membrane tubules before their fission from the Golgi apparatus (*Miserey-Lenkei et al., 2010*). While our data indicate that such contribution is not essential for the formation of secretory carriers, we did find that Rab6 vesicles generated in the presence of KIF13B alone were smaller than those formed when KIF5B was also expressed. As indicated above, KIF5B detaches less readily under load than KIF13B (*Arpağ et al., 2019*; *Arpağ et al., 2014*), and could be more efficient in pulling out a longer membrane tube before its fission. We note, however, that the cell area within the immediate vicinity of the Golgi is not visualized optimally by TIRF microscopy and was not included in our analysis. Therefore, additional imaging experiments and analysis would be needed to address the mechanisms of vesicle exit from the Golgi, the contribution of kinesins to this process and the timing of the recruitment of the two kinesins during vesicle movement from the Golgi to the cell periphery. Ideally, this should be done with fluorescently tagged motors expressed from the endogenous locus.

An obvious function of kinesins in secretory vesicle transport, confirmed by our work, is the control of the spatial distribution of exocytosis events. In HeLa cells, Rab6 vesicles preferentially fuse with the plasma membrane at the cell periphery in the vicinity of focal adhesions (*Fourriere et al., 2019*; *Grigoriev et al., 2007*; *Stehbens et al., 2014*). Exocytosis preferentially occurs at the sites where dynamic MT plus ends are attached to the cell cortex by the complex of CLASPs, LL5β, KANK1 and ELKS, which together with MICAL3 regulates vesicle fusion (*Grigoriev et al., 2011*; *Lansbergen et al., 2006*). Such spatial organization can contribute to cell migration by directing secretion of new membrane and extracellular matrix proteins to the leading edge of the cell and also by controlling focal adhesion turnover (*Schmoranzer et al., 2000*; *Stehbens et al., 2014*).

Our data suggest that KIF13B, being a minor player in Rab6 vesicle transport in HeLa cells, likely due to its low abundance, nevertheless contributes to the spatial organization of exocytosis by bringing the vesicles to the tips of dynamic MTs. Cargo unloading at dynamic MT ends could be a general property of kinesin-3 family members, as it was also recently proposed for KIF1A (*Guedes-Dias et al., 2019*). KIF5B, which strongly dominates Rab6 vesicle transport, is in contrast restricted to the older, more stable MT population (*Cai et al., 2009*; *Guardia et al., 2016*; *Tas et al., 2017*). The underlying mechanisms of kinesin-1 selectivity are complex, and might involve the ability of the motor to regulate the properties of the tracks on which it walks (*Shima et al., 2018*). Importantly,

KIF5B shows a very strong dependence on MAP7 family members, which control both kinesin recruitment to the MT and its activation (*Hooikaas et al., 2019*; *Metzger et al., 2012*; *Monroy et al., 2018*; *Pan et al., 2019*). MAP7 and its homologues are more abundant on older MTs, apparently because of their slow association with growing MTs. In COS7 cells, where MTs are more dynamic than in the HeLa cells studied here, overexpressed GFP-MAP7 was reported to form a sharp boundary along the MT shaft, associated with the end of the stable MT segment (*Tymanskyj and Ma, 2019*). It can be expected that in cells with highly dynamic MTs, Rab6 transport to MT plus ends will be more dependent on kinesin-3 family members. Interestingly, since KIF13B expression is able to rescue Rab6 vesicle transport in the absence of KIF5B, and the distribution of MAP7 family members is not affected by KIF5B loss, it appears that KIF13B can work well not only on freshly polymerized but also on older, MAP7-positive MTs, even though the motility of kinesin-3 has been shown to be negatively affected by MAP7 (*Monroy et al., 2020*). The expression of MAP7 proteins in HeLa cells is thus sufficient to support transport by kinesin-1 but not to hinder KIF13B activity. Alternatively, additional MAPs promoting kinesin-3 motility may allow KIF13B to move on MTs in spite of the presence of MAP7 family members.

To summarize, our work demonstrates how the cooperation of two different kinesins promotes efficient cargo transport along complex MT networks. Furthermore, our study shows that direct visualization of motors on moving cargo, which in the future could be extended to cargo adaptors, provides insight into the motor activity and can be used to study the relationships between different players during bi-directional multimotor MT-dependent transport.

## Materials and methods

### Constructs, antibodies, and reagents

GFP-Rab6A construct (*Matanis et al., 2002*), GFP-Eg5 (*Jiang et al., 2012*), pßactin-PEX3-mRFP and KIF5B(1-807)-GFP-FRB (*Kapitein et al., 2010b*), GFP-Rab11 (*Hoogenraad et al., 2010*), NPY-GFP (*Schlager et al., 2010*), KIF13B(1-444)-GFP-FRB (*Lipka et al., 2016*), KIF13A-GFP (*Schou et al., 2017*) and FKBP-mCherry-Rab6A (*Schlager et al., 2014*) were described previously. The addition of FKBP to the N-terminus of mCherry in the mCherry-Rab6 fusion had no detectable effect on the behavior of this marker (*Schlager et al., 2014*), and it is termed mCherry-Rab6A throughout the paper. GFP-KIF13B was a gift from Dr. Athar Chishti (University of Illinois College of Medicine, Chicago, USA) and used as a template for GFP-KIF13B deletion constructs prepared by PCR-based strategy. PCR products were sub-cloned in pEGFP expression vectors (Clontech). KIF13B was amplified by PCR from GFP-KIF13B and inserted into a Bio-mCherry-C1 vector to make mCherry-KIF13B. Expression constructs for KIF5B, KIF1B and KIF1C were obtained by inserting the corresponding human cDNA (image clone 8991997 and KIAA1448 and KIAA0706 cDNAs, a gift from Kazusa DNA Research Institute Foundation *Kikuno, 2004*) into pßactin-GFP vector (KIF5B) or pEGFP vector (Clontech) with excised GFP (KIF1B and KIF1C) by PCR-based strategies.

PAUF-mRFP (*Wakana et al., 2012*) was a gift from Dr. Vivek Malhotra (Centre for Genomic Regulation, Barcelona, Spain), TagRFP-T-Rab6A a gift from Dr. Yuko Mimori-Kiyosue (RIKEN Center for Developmental Biology, Japan) and streptavidin-KDEL-SBP-GFP-E-Cadherin and streptavidin-KDEL-solubleGFP-SBP (*Boncompain et al., 2012*) were a gift from Dr. Franck Perez (Institut Curie, Paris, France).

The following antibodies were used in this study: mouse monoclonal antibodies against Rab6 (*Schiedel et al., 1995*) and Ku80 (BD Biosciences, Cat#611360, RRID:AB_398882), mouse polyclonal antibody against MAP7 (Abnova, Cat#H00009053-B01P, RRID:AB_10714227), rabbit polyclonal antibodies against KIF1B (Bethyl, Cat#A301-055A, RRID:AB_2131416), KIF1C (Cytoskeleton, Cat# AKIN11-A, RRID:AB_10708792), KIF5B/UKHC (Santa Cruz Biothnology, Cat# SC28538, clone H50, RRID:AB_2280915), Eg5 (Abcam, Cat#ab61199, RRID:AB_941397), KLC-1 (Santa Cruz Biotechnology, Cat# sc25735, clone H75, RRID:AB_2280879), MAP7D1, (Atlas Antibodies, Cat# HPA028075, RRID:AB_10603778), MAP7D3 (Atlas Antibodies, Cat# HPA035598, RRID:AB_10671108) and GFP (Abcam, Cat# ab290, RRID:AB_303395), and rat monoclonal antibody against α-tubulin YL1/2 (Abcam, Cat#ab6160, RRID:AB_305328). The anti-KIF13B polyclonal antibody was produced by immunizing rabbits with a purified GST-KIF13B protein (amino acids 1096–1143) expressed in BL21

*E. coli* using the pGEX-5X-3 vector (GE Healthcare). The antiserum was affinity purified using the antigen coupled to Dyna M-280 Streptavidin beads (Life Technologies).

For immunofluorescence and flow cytometry experiments we used Alexa488-, Alexa568- and Alexa-641 conjugated goat secondary antibodies (Invitrogen/Molecular probes). For Western blotting we used IRDye 680LT and 800CW goat anti-mouse and anti-rabbit antibodies (Li-Cor Biosciences).

Thymidine and biotin were from Sigma-Aldrich and rapalog was from Clontech.

## Cell culture and treatment

HeLa (Kyoto) and MRC5 cell lines were cultured in DMEM/Ham's F10 medium (50/50%) supplemented with 10% fetal calf serum, penicillin and streptomycin and were routinely checked for mycoplasma contamination (LT07-518 Mycoalert assay, Lonza).

Plasmid transfection was performed using FuGENE 6 (Promega) according to the manufacturers' protocol 24 hr before experiments whereas HiPerfect (Qiagen) reagent was used for transfecting 10 nM siRNA per target 72 hr before experiments. The siRNAs used in this study were synthesized by Sigma and were directed against the following sequences:

> Control/Luciferase: 5'-CGTACGCGGAATACTTCGA-3';
> Eg5: 5'-GAGCCCAGATCAACCTTTA-3'
> MAP7D1: 5'-TCATGAAGAGGACTCGGAA-3'
> MAP7D3: 5'-AACCTACATTCGTCTACTGAT-3'

For Eg5-related experiments, HeLa cells were treated with 2 mM thymidine 12 hr after transfection (with control and Eg5 siRNA) until the end of the experiment.

HeLa knockout lines were generated using the CRISPR/Cas9 method (*Ran et al., 2013*) by transfecting pSpCas9-2A-Puro (Addgene) vectors containing the following targeting sequences for gRNAs:

> KIF13B: 5'-TGCGGATACGACCCATGAAC-3'
> KIF5B: 5'-TTCACTTCAGACTCGTTGAG-3'
> KIF1B: 5'-TCAGCTTCGACTATTCCTAC-3'
> KIF1C: 5'-GCTGGTCTCACGGGCGTTAA-3'
> MAP7: 5'-CGCCCTGCCTCTGCAATTTC-3'

One day after transfection, cells were treated with 2 µg/ml puromycin for 2 days and allowed to recover for 5 days. Selected cells were then diluted in 96-wells plates for growing single cell colonies that were tested by Western blotting. Alternatively, the polyclonal population obtained right after puromycin selection and recovery was tested by Western blotting, aliquots were frozen and used within 2 months after thawing (KO mixed population).

For RUSH assays (*Boncompain et al., 2012*), 24 hr after transfection with Streptavidin-KDEL-SBP-GFP-E-Cadherin and Streptavidin-KDEL-solubleGFP-SBP, HeLa cells were treated with 40 µM biotin for synchronization of secretion before live imaging or flow cytometry analysis.

For inducible peroxisome trafficking assay (*Kapitein et al., 2010b*), MRC5 cells were treated with 100 nM rapalog for kinesin recruitment before live imaging, which was performed ~10–40 min later.

## Western blotting

Extracts of HeLa cells were prepared in RIPA buffer (10 mM Tris-HCl pH 8, 140 mM NaCl, 1 mM EDTA, 1 mM EGTA, 1% Triton X-100, 0.1% SDS, protease inhibitor cocktail (Complete - Sigma)). SDS-PAGE and Western blot analysis were performed according to standard procedures and developed with the Odyssey Infrared Imaging system (Li-Cor Biosciences).

## Immunofluorescence cell staining

HeLa cells were fixed in 4% paraformaldehyde for 10 min at room temperature or with 100% methanol for 10 min at −20℃ for tubulin staining, permeabilized with 0.15% Triton X-100 in phosphate buffered saline (PBS) for 5 min, blocked in 2% bovine serum albumin/0.07% Tween 20 in PBS and sequentially incubated with primary and secondary antibodies in blocking buffer for 1 hr at room temperature. Cells were washed three times with 0.07% Tween 20 in PBS after each labeling step.

Slides were then air-dried and mounted in Vectashield mounting medium, which in some cases was supplemented with DAPI (Vector laboratories).

## Flow cytometry

After biotin treatment for the indicated time, HeLa cells were harvested and surface immunostaining was performed on ice. Cells were labeled with anti-GFP antibody for 45 min, fixed with 2% paraformaldehyde for 10 min, washed with PBS, incubated with Alexa641-conjugated secondary antibody, washed with PBS again and analyzed using a FACS Canto Flow Cytometry System (BD Biosciences). Viable cells were selected using FSC and SSC parameters, and signals for 'cellular' GFP (intrinsic GFP signal) and 'surface-expressed' GFP (from staining with the Alexa641-conjugated secondary antibodies) were collected per cell. Cells positive for E-cadherin expression were gated using the GFP channel and the geometric mean of the Alexa641 signal intensity was calculated for this population. This value was quantified before and 1 hr after biotin treatment and the fold change was calculated. The same procedure was applied to quantify the decrease in soluble-GFP cellular content using the GFP signal without immunostaining.

## Image acquisition

Images of fixed cells were collected with a wide-field fluorescence microscope Nikon Eclipse 80i equipped with C-HGFI Fiber Illuminator 'Intensilight' (Nikon), Plan Apo VC 100x N.A. 1.40 oil objective (Nikon) and ET-DAPI (49000, Chroma), ET-GFP (49002, Chroma), ET-mCherry (49008, Chroma) and ET-GFP/mCherry (59022, Chroma) filters and a CoolSNAP HQ2 CCD camera (Photometrics) or with a confocal fluorescence microscope LSM 700 (Zeiss) equipped with a Plan-Apochromat 63x/ 1.40 (oil) objective (Zeiss).

For live cell imaging Total Internal Reflection Fluorescence microscopy (TIRFM) and Spinning Disc confocal fluorescent microscopy were used.

TIRFM was performed on an inverted research microscope Nikon Eclipse Ti-E (Nikon), equipped with the perfect focus system (Nikon), Nikon Apo TIRF 100x N.A. 1.49 oil objective (Nikon) and Nikon TIRF-E motorized TIRF illuminator modified by Roper Scientific France/PICT-IBiSA, Institut Curie (currently Gataca Systems). The system was also equipped with ASI motorized stage MS-2000-XY (ASI), Photometrics Evolve 512 EMCCD camera (Photometrics) and controlled by the MetaMorph 7.8 software (Molecular Devices). 491 nm 100 mW Calypso (Cobolt) and 561 nm 100 mW Jive (Cobolt) lasers were used as the light sources. We used ET-GFP filter set (49002, Chroma) for imaging of proteins tagged with GFP; ET-mCherry filter set (49008, Chroma) for imaging of proteins tagged with mCherry; for simultaneous imaging of green and red fluorescence we used ET-GFP/ mCherry filter set (59022, Chroma) together with Optosplit III beamsplitter (Cairn Research Ltd, UK) equipped with double emission filter cube configured with ET525/50 m, ET630/75 m and T585LPXR (Chroma). 16-bit images were projected onto the EMCCD chip with intermediate lens 2.5X (Nikon C mount adapter 2.5X) at a magnification of 0.063 µm/pixel. To keep cells at 37°C we used stage top incubator (model INUBG2E-ZILCS, Tokai Hit).

Azimuthal spinning TIRFM was performed on an inverted research microscope Nikon Eclipse Ti-E (Nikon), equipped with the perfect focus system (Nikon), Nikon Apo TIRF 100x N.A. 1.49 oil objective (Nikon) and iLas2 system (Dual Laser illuminator for azimuthal spinning TIRF (or Hilo) illumination and Simultaneous Targeted Laser Action) from Roper Scientific (Evry, France; now Gataca Systems) with a custom modification for targeted Photoablation using a 532 nm pulsed laser. The system was also equipped with ASI motorized stage MS-2000-XY (ASI), Photometrics Evolve Delta 512 EMCCD camera (Photometrics) and controlled by the MetaMorph 7.8 software (Molecular Devices). Stradus 405 nm (100 mW, Vortran), Stradus 488 nm (150 mW, Vortran) and OBIS 561 nm (100 mW, Coherent) lasers were used as the light sources. We used ET-GFP filter set (49002, Chroma) for imaging of proteins tagged with GFP; ET-mCherry filter set (49008, Chroma) for imaging of proteins tagged with mCherry; for simultaneous imaging of green and red fluorescence we used ET-GFP/mCherry filter set (59022, Chroma) together with Optosplit III beamsplitter (Cairn Research Ltd, UK) equipped with double emission filter cube configured with ET525/50 m, ET630/75 m and T585LPXR (Chroma). For simultaneous imaging of blue, green and red fluorescence we used ZT405/488/561/640rpc ZET405/488/561/635 m filter set (TRF89901, Chroma) together with Optosplit III beamsplitter (Cairn Research Ltd, UK) equipped with triple emission filter cube configured with ET460/50 m, ET525/50

m, ET630/75 m, T495LPXR and T585LPXR (Chroma). 16-bit images were projected onto the EMCCD chip with intermediate lens 2.5X (Nikon C mount adapter 2.5X) at a magnification of 0.065 µm/pixel. To keep cells at 37°C we used stage top incubator (model INUBG2E-ZILCS, Tokai Hit).

For time-lapse fluorescence imaging of GFP-E-Cadherin, images were collected with spinning disk confocal microscopy on inverted research microscope Nikon Eclipse Ti-E (Nikon), equipped with the perfect focus system (Nikon) Nikon Plan Apo 60x N.A. 1.40 oil objective (Nikon) and a spinning disk-based confocal scanner unit (CSU-X1-A1, Yokogawa). The system was also equipped with ASI motorized stage with the piezo plate MS-2000-XYZ (ASI), Photometrics Evolve Delta 512 EMCCD camera (Photometrics) and controlled by the MetaMorph 7.8 software (Molecular Devices). 491 nm 100 mW Calypso (Cobolt) laser was used as the light sources. We used ET-GFP filter set (49002, Chroma) for imaging of proteins tagged with GFP; 16-bit images were projected onto the EMCCD chip with intermediate lens 2.0X (Edmund Optics) at a magnification of 111 µm/pixel. To keep cells at 37°C we used stage top incubator (model INUBG2E-ZILCS, Tokai Hit).

Fluorescence Recovery After Photobleaching (FRAP) experiments have been performed on either TIRF or Spinning Disc microscopes, equipped with iLas or iLas2 platforms and using Targeted Laser Action options of iLas or iLas2 and controlled with iLas/iLas2 softwares (Roper Scientific, now Gataca Systems).

## Image analysis

To prepare images for publication, we used ImageJ to perform adjustments of levels and contrast and generate maximum intensity projections. Two-color-intensity profiles along a line were made using the 'plot profile' option in ImageJ for each channel.

For manual analysis, kymographs of Rab6 and kinesin tracks were made using MetaMorph 7.8 software (Kymograph option). Kymographs have been analyzed in MetaMorph software. Two parameters have been measured for the segments of kymograph representing particle displacements, the angle and the length, and these parameters were used to calculate the speed of movement in SigmaPlot software.

The distribution of MAP7 and $\alpha$-tubulin signals along the cell radius was determined using the 'Radial Profile Angle' plugin of ImageJ with an integration angle of 45°. Each intensity profile was then normalized according to: $\frac{x - x_{min}}{x_{max} - x_{min}}$ and the profile of the ratio between the normalized values for MAP7 and $\alpha$-tubulin was calculated per cell.

For analyzing the distribution Rab11-positive endosomes, the ratio between the mean signal intensity in the perinuclear region (obtained by scaling-down the cell outline by 25%) and in the whole cell was calculated using maximum intensity projections over time of GFP-Rab11 signal and expressed in percent.

Kinesin-GFP fluorescence intensities were analyzed by measuring the mean GFP intensity in circles of 7 pixels in diameter centered on the GFP dots and correcting it for the mean intensity in circles of the same size positioned just next to the fluorescent spot. For full-length kinesins, only GFP spots colocalizing with Rab6 vesicles were selected.

The NPY exocytosis events were analyzed by visually inspecting TIRF imaging of NPY-GFP and looking for a fast burst of fluorescence followed by the disappearance of the signal and confirming it by kymograph analysis. The position of NPY exocytosis was determined by measuring the distance between the exocytosis spot and the center of the Golgi region (determined based on the morphology of NPY signal).

## Automatic tracking of Rab6 vesicles

TIRFM images of Rab6 vesicles were analyzed using ImageJ software with the recently developed SOS plugin (*Yao et al., 2017*), where we combined 'SOS detector 3D module' as particle detector and 'SOS linker (NGMA) module' for particle linking. MTJ (MTrackJ; *Rueden et al., 2015*; https://github.com/imagescience/MTrackJ/; copy archived at swh:1:rev:3e712c734d229e2fbe46627b711-b74e83ba81c1f) Simple Track Segment module was applied to the resulting tracks to find segments of directional movement (runs) inside each trajectory. Golgi area was manually excluded from the analysis. For this analysis, only tracks of 20 or more frames were considered. A segment was assigned to be a directional run if the cosine of angle between all two consecutive velocity vectors was above 0.6 and it lasted more than six frames. If inside such a segment there were more than

three consecutive displacements of one pixel or less per frame, at this point the segment was further split into two. Duration and length of each directional run was extracted and its speed was calculated using the complementary SAID plugin (*Yao et al., 2017*). We recorded the location of each run inside the corresponding track and used this information to calculate statistics on fractions of processive movement per track or duration (*Figure 2G,H,I*). The raw tracks, extracted runs and MATLAB code for this analysis are available at https://doi.org/10.6084/m9.figshare.c.5177636.v1.

The resulting speeds were expressed in two ways: either the mean of the speed was calculated per cell or a frequency distribution of the speed of all individual runs was built.

The fitting of the distribution of Rab6 velocity with either one Gaussian or the sum of two Gaussian curves, as well as their comparison using extra sum-of-squares F test method were performed using GraphPad Prism5.

## Analysis of kinesin distribution on moving vesicles

### Detection

For simultaneous two-color imaging of vesicles labeled with Rab6, motor proteins and PAUF, we used OptoSplit III beamsplitter (Cairn Research Ltd, UK) equipped with double emission filter cube projecting two channels on the camera chip simultaneously. To account for chromatic aberrations of the objective, images of a calibration photomask with round 500 nm features positioned equidistantly at 2 µm distance (Compugraphics, UK) were acquired simultaneously in GFP and mCherry channels using transmitted bright-field illumination using the procedures described in *Aher et al., 2018*; *Maurer et al., 2014*. Based on feature detections we made sub-pixel channel alignment and non-linear registration using B-spline transform implemented in our Detection of Molecules ImageJ plugin and described earlier (version 1.1.6, https://github.com/ekatrukha/DoM_Utrecht, *Chazeau et al., 2016*). The plugin provided sub-pixel localized coordinates of spots in the 'green' channel and corrected for chromatic aberration coordinates of spots in the 'red' channel.

### Colocalization and tracking

To determine colocalization, for each time frame we imported 'green' coordinates as frame one and 'red' coordinates as frame two to MATLAB (MathWorks, 2011b). We used SimpleTracker routine by Jean-Yves Tinevez (https://www.github.com/tinevez/simpletracker; *Jean-Yves, 2019*), implementing nearest neighbor search with a distance threshold of 8 pixels (0.52 µm) to assemble 'short' two frame tracks. Linked detections were considered colocalized, and for each detection average $x$ and $y$ coordinates between two channels were calculated and stored. For each colocalization detection, we kept records of the original coordinates in each channel. On the next step, we used those averaged coordinates to assemble tracks of colocalized particles over the duration of the whole movie using the same SimpleTracker routine with a distance threshold of 10 pixels (0.65 µm) and the maximum number of gap closing frames equal to 5. From those tracks we extracted segments of persistent directional movement (runs) as described earlier (*Katrukha et al., 2017*). In short, for each track, an array of instant velocity vectors was generated as a difference between two positions of a vesicle in two consecutive frames divided by the time between frames. A cosine of the angle between two consecutive velocity vectors was used as a directional correlation measure. For every trajectory we filtered segments where the value of cosine was above defined threshold. To find runs we used the lower threshold value of 0.6, corresponding to an approximately 100° cone facing in the direction of movement. Only runs longer than 0.3 s and with displacement above 0.5 µm were considered. For the directional filtering of GFP-KIF13B/KIF5B-GFP and mCherry-Rab6A pairs, we used coordinates of mCherry-Rab6A. For PAUF-mRFP and GFP-Rab6A pair we used GFP-Rab6A, and for KIF5B-GFP and mCherry-KIF13B pair we used averaged coordinates. The projected distance was calculated as a dot product between the velocity vector and the vector from the cargo to the motor, divided by the length of the velocity vector.

### Analysis of opposite polarity runs

For each directional movement segment with colocalizing signals of the two markers, we first calculated the average velocity vector $v_i$, coordinates of the center of mass and the average value of the projected distance. For trajectories with more than one run we listed all possible different run pairs. For each pair we calculated the cosine between average velocity vectors. If the cosine value was

below −0.6 (i.e. angle between the velocity vectors was larger than 125 degrees), we considered these runs to have opposite polarity. As an additional constraint, we only considered pairs where the distance between the centers of mass was smaller than 1 µm. The run with the larger average projected distance was considered the 'forward' run, while the other one was assigned to be the 'backward' run.

All raw and segmented tracks and corresponding MATLAB source code used for analysis are available at https://doi.org/10.6084/m9.figshare.c.5177636.v1.

## FRAP analysis

To measure the KIF13B turnover on moving Rab6 vesicles, GFP-Rab6A stably expressing cells were transfected with mCherry-KIF13B. We used Targeted Laser Action option of iLas platform (Roper Scientific, now Gataca Systems) installed on Nikon-based TIRF microscope. Using FRAP-on-Fly option of iLas software we photobleached mCherry-KIF13B with a 561 laser (100%) in a ~ 0.5 µm circle area surrounding a moving vesicle. Keeping unaffected green signal of GFP-Rab6A as a reference, we measured the intensities of both GFP-Rab6A and mCherry-KIF13B in ~0.5 µm circle area of a vesicle over time after bleaching. We subtracted the background (measured in the ~0.5 µm circle area next to the vesicle). Next, we normalized the measured intensities to the average of intensities at 10 time points before photobleaching. We then averaged all measured vesicles.

## Statistics

Statistical analyses were performed using GraphPad Prism five and significance was assessed using Mann-Whitney U test. The sample size is indicated in the figure legends. All graphs show mean ± SD, except in *Figure 1H*, *Figure 5G, H, M* that depict mean ± SEM. No explicit power analysis was used to determine sample size and no masking was used for the analysis.

## Acknowledgements

We thank V Malhotra, F Perez, Y Mimori-Kiyosue, A Chishti, F Perez and Kazusa DNA Research Institute Foundation for the gift of materials, and the members of the Dumont lab (Department of Bioengineering and Therapeutic Sciences, UCSF, USA) and the Chang and Wittman labs (Department of Cell and Tissue Biology, UCSF, USA) for helpful discussions.

This work was supported by the European Research Council (ERC) Synergy grant 609822 and Netherlands Organization for Scientific Research ALW Open Program grant 824.15.017 to AA, the Marie Curie IEF fellowships to MM, a Netherlands Organization for Scientific Research STW grant OTP13391 to EM and AA, and a Fundação para a Ciência e a Tecnologia fellowship to AS-M, and the ERC Consolidator grant 819219 to LCK.

## Additional information

### Competing interests

Lotte B Pedersen: Reviewing editor, *eLife*. Anna Akhmanova: Deputy editor, *eLife*. The other authors declare that no competing interests exist.

### Funding

| Funder | Grant reference number | Author |
| --- | --- | --- |
| H2020 European Research Council | Synergy grant 609822 | Anna Akhmanova |
| H2020 European Research Council | Consolidator grant 819219 | Lukas C Kapitein |
| Nederlandse Organisatie voor Wetenschappelijk Onderzoek | ALW Open Program grant 824.15.017 | Anna Akhmanova |
| H2020 Marie Skłodowska-Curie Actions | IEF fellowship | Maud Martin |
| Nederlandse Organisatie voor | STW grant OTP13391 | Erik Meijering |

Wetenschappelijk Onderzoek

| Fundação para a Ciência e a Tecnologia | PhD fellowship | Andrea Serra-Marques |

The funders had no role in study design, data collection and interpretation, or the decision to submit the work for publication.

### Author contributions

Andrea Serra-Marques, Conceptualization, Resources, Formal analysis, Funding acquisition, Validation, Investigation, Visualization, Methodology, Writing - original draft, Writing - review and editing; Maud Martin, Conceptualization, Formal analysis, Funding acquisition, Validation, Investigation, Visualization, Methodology, Writing - original draft, Writing - review and editing; Eugene A Katrukha, Conceptualization, Software, Formal analysis, Investigation, Visualization, Methodology, Writing - original draft, Writing - review and editing; Ilya Grigoriev, Formal analysis, Validation, Investigation, Visualization, Methodology, Writing - original draft; Cathelijn AE Peeters, Veronika Solianova, Investigation; Qingyang Liu, Peter Jan Hooikaas, Resources, Investigation; Yao Yao, Software; Ihor Smal, Software, Supervision; Lotte B Pedersen, Resources; Erik Meijering, Software, Supervision, Funding acquisition; Lukas C Kapitein, Supervision, Funding acquisition, Methodology, Writing - original draft; Anna Akhmanova, Conceptualization, Supervision, Funding acquisition, Investigation, Writing - original draft, Project administration, Writing - review and editing

### Author ORCIDs

Andrea Serra-Marques  https://orcid.org/0000-0003-4215-3024
Maud Martin  https://orcid.org/0000-0003-0048-6437
Peter Jan Hooikaas  https://orcid.org/0000-0001-9849-9193
Lotte B Pedersen  https://orcid.org/0000-0002-9749-3758
Erik Meijering  https://orcid.org/0000-0001-8015-8358
Lukas C Kapitein  https://orcid.org/0000-0001-9418-6739
Anna Akhmanova  https://orcid.org/0000-0002-9048-8614

### Decision letter and Author response

Decision letter https://doi.org/10.7554/eLife.61302.sa1
Author response https://doi.org/10.7554/eLife.61302.sa2

## Additional files

### Supplementary files

• Transparent reporting form

### Data availability

The source data that support the conclusions of the paper are included as supplementary files for all figures containing plots (all 8 Main figures and Figure Supplements to Figs 1, 2, 3, 5 and 8). The custom software used for movement tracking and analysis in this manuscript can be found at https://github.com/imagescience/MTrackJ/ (copy archived at https://archive.softwareheritage.org/swh:1:rev:3e712c734d229e2fbe46627b711b74e83ba81c1f/). All raw and segmented trajectories and corresponding custom source code are available at https://doi.org/10.6084/m9.figshare.c.5177636.v1.

The following dataset was generated:

| Author(s) | Year | Dataset title | Dataset URL | Database and Identifier |
|---|---|---|---|---|
| Liu Q, Hooikaas PJ, Yao Y, Solianova V, Smal I, Pedersen LB, Meijering E, Kapitein LC, Akhmanova A, Serra- | 2020 | Concerted action of kinesin-1 KIF5B and kinesin-3 KIF13B promotes efficient transport of exocytotic vesicles to microtubule plus ends | https://doi.org/10.6084/m9.figshare.c.5177636.v1 | figshare, 10.6084/m9.figshare.c.5177636.v1 |

Marques A, Martin M, Katrukha EA, Grigoriev I, Peeters CAE

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

# Appendix

**Appendix 1—key resources table**

| Reagent type (species) or resource | Designation | Source or reference | Identifiers | Additional information |
|---|---|---|---|---|
| Cell line (*Homo sapiens*) | HeLa S3 (Kyoto) | ATCC | CCL-2.2 | |
| Cell line (*Homo sapiens*) | MRC5 | ATCC | CCL-171 | |
| Cell line (*Homo sapiens*) | KIF5B-KO HeLa | This paper | | CRISPR/Cas9 generated monoclonal HeLa line |
| Cell line (*Homo sapiens*) | KIF5B/KIF13B-KO HeLa | This paper | | CRISPR/Cas9 generated monoclonal HeLa line |
| Cell line (*Homo sapiens*) | 4X (KIF5B/KIF13B/KIF1B/KIF1C)-KO HeLa | This paper | | CRISPR/Cas9 generated monoclonal HeLa line |
| Cell line (*Homo sapiens*) | MAP7-KO HeLa | *Hooikaas et al., 2019*; PMID:30770434 | | |
| Transfected construct (*Homo sapiens*) | GFP-Rab6A | *Matanis et al., 2002*; PMID:12447383 | | Expression construct transfected in HeLa |
| Transfected construct (*Homo sapiens*) | GFP-Eg5 | *Jiang et al., 2012*; PMID:22885064 | | Expression construct transfected in HeLa |
| Transfected construct (*Homo sapiens*) | pβactin-PEX3-mRFP | *Kapitein et al., 2010b*; PMID:20923648 | | Expression construct transfected in MRC5/peroxisome trafficking assay |
| Transfected construct (*Homo sapiens*) | KIF5B(1-807)-GFP-FRB | *Kapitein et al., 2010b*; PMID:20923648 | | Expression construct transfected in MRC5/peroxisome trafficking assay |
| Transfected construct (*Homo sapiens*) | GFP-Rab11 | *Hoogenraad et al., 2010*; PMID:21057633 | | Expression construct transfected in HeLa |
| Transfected construct (*Homo sapiens*) | NPY-GFP | *Schlager et al., 2010*; PMID:20360680 | | Expression construct transfected in HeLa |
| Transfected construct (*Homo sapiens*) | KIF13B(1-444)-GFP-FRB | *Lipka et al., 2016*; PMID:26758546 | | Expression construct transfected in MRC5/peroxisome trafficking assay |
| Transfected construct (*Homo sapiens*) | KIF13A-GFP | *Schou et al., 2017*; PMID:28134340 | | Expression construct transfected in HeLa |

*Continued on next page*

*Appendix 1—key resources table continued*

| Reagent type (species) or resource | Designation | Source or reference | Identifiers | Additional information |
|---|---|---|---|---|
| Transfected construct (*Homo sapiens*) | FKBP-mCherry-Rab6A | *Schlager et al., 2014*; PMID:25176647 | | Expression construct transfected in HeLa, termed mCherry-Rab6A in this paper |
| Transfected construct (*Homo sapiens*) | TagBFP-Rab6A | This paper | | Expression construct transfected in HeLa |
| Transfected construct (*Homo sapiens*) | GFP-KIF13B | gift from gift from Dr. Athar Chishti (University of Illinois College of Medicine, Chicago, USA) *Venkateswarlu et al., 2005*; PMID:15923660 | | Expression construct transfected in HeLa |
| Transfected construct (*Homo sapiens*) | TagRFP-T-Rab6A | gift from Dr. Yuko Mimori-Kiyosue (RIKEN Center for Developmental Biology, Japan) | | Expression construct transfected in HeLa |
| Transfected construct (*Homo sapiens*) | GFP-KIF13B | This paper | | Expression construct transfected in HeLa |
| Transfected construct (*Homo sapiens*) | GFP-KIF13B Δ motor (393–1826) | This paper | | Expression construct transfected in HeLa |
| Transfected construct (*Homo sapiens*) | GFP-KIF13B-FHA/MBS (364–1013) | This paper | | Expression construct transfected in HeLa |
| Transfected construct (*Homo sapiens*) | GFP-KIF13B C1 (440–1826) | This paper | | Expression construct transfected in HeLa |
| Transfected construct (*Homo sapiens*) | GFP-KIF13B C2 (607–1826) | This paper | | Expression construct transfected in HeLa |
| Transfected construct (*Homo sapiens*) | GFP-KIF13B C3 (752–1826) | This paper | | Expression construct transfected in HeLa |
| Transfected construct (*Homo sapiens*) | GFP-KIF13B C4 (1014–1826) | This paper | | Expression construct transfected in HeLa |
| Transfected construct (*Homo sapiens*) | GFP-KIF13B C5 (607–1623) | This paper | | Expression construct transfected in HeLa |
| Transfected construct (*Homo sapiens*) | GFP-KIF13B C6 (993–1292) | This paper | | Expression construct transfected in HeLa |
| Transfected construct (*Homo sapiens*) | GFP-KIF13B-CAP-Gly (1623–1826) | This paper | | Expression construct transfected in HeLa |
| Transfected construct (*Homo sapiens*) | mCherry-KIF13B | This paper | | Expression construct transfected in HeLa |

*Continued on next page*

*Appendix 1—key resources table continued*

| Reagent type (species) or resource | Designation | Source or reference | Identifiers | Additional information |
|---|---|---|---|---|
| Transfected construct (*Homo sapiens*) | KIF5B-GFP | This paper | | Expression construct transfected in HeLa |
| Transfected construct (*Homo sapiens*) | KIF1B | This paper | | Expression construct transfected in HeLa |
| Transfected construct (*Homo sapiens*) | KIF1C | This paper | | Expression construct transfected in HeLa |
| Transfected construct (*Homo sapiens*) | PAUF-mRFP | *Wakana et al., 2012*; PMID:22909819 | | Expression construct transfected in HeLa |
| Transfected construct (*Homo sapiens*) | streptavidin-KDEL-SBP-GFP-E-Cadherin | *Boncompain et al., 2012*; PMID:22406856 | | Expression construct transfected in HeLa/RUSH assay |
| Transfected construct (*Homo sapiens*) | streptavidin-KDEL-solubleGFP-SBP | *Boncompain et al., 2012*; PMID:22406856 | | Expression construct transfected in HeLa/RUSH assay |
| Antibody | anti-Rab6 (Mouse monoclonal) | *Schiedel et al., 1995*; PMID:8521955 | | IF (1:300) |
| Antibody | anti-Ku80 (Mouse monoclonal) | BD Biosciences | Cat#:611360, RRID:AB_398882 | WB (1:2000) |
| Antibody | anti-EB1 (Mouse monoclonal) | BD Biosciences | Cat#:610252; RRID:AB_2276073 | IF (1:200) |
| Antibody | anti-EEA1 (Mouse monoclonal) | BD Biosciences | Cat#:610456, RRID:AB_397829 | IF (1:100) |
| Antibody | anti-MAP7 (Mouse polyclonal) | Abnova | Cat#:H00009053-B01P, RRID:AB_10714227 | IF (1:300) WB (1:1000) |
| Antibody | anti-KIF1B (Rabbit polyclonal) | Bethyl | Cat#:A301-055A, RRID:AB_2131416 | WB (1:500) |
| Antibody | anti-KIF1C (Rabbit polyclonal) | Cytoskeleton | Cat#:AKIN11-A, RRID:AB_10708792 | WB (1:300) |
| Antibody | Anti-KIF5B/UKHC (Rabbit polyclonal) | Santa Cruz Biotechnology | Cat#:SC28538, clone H50, RRID:AB_2280915 | WB (1:1000) |
| Antibody | anti-Eg5 (Rabbit polyclonal) | Abcam | Cat#:ab6119, RRID:AB_941397 | WB (1:500) |
| Antibody | anti-KLC1 (Rabbit polyclonal) | Santa Cruz Biotechnology | Cat#:sc25735, clone H75, RRID:AB_2280879 | IF (1:200) WB (1:1000) |
| Antibody | Anti-MAP7D1 (Rabbit polyclonal) | Atlas antibodies | Cat#:HPA028075, RRID:AB_10603778 | IF (1:300) WB (1:1000) |

*Continued on next page*

*Appendix 1—key resources table continued*

| Reagent type (species) or resource | Designation | Source or reference | Identifiers | Additional information |
|---|---|---|---|---|
| Antibody | anti-Map7D3 (Rabbit polyclonal) | Atlas antibodies | Cat#:HPA035598, RRID:AB_10671108 | IF (1:300) WB (1:1000) |
| Antibody | Anti-GFP (Rabbit polyclonal) | Abcam | Cat#: ab290, RRID:AB_303395 | FACS (1:500) |
| Antibody | Anti-FAK (Phospho-Tyr397) (Rabbit polyclonal) | Biosource | Cat#:MBS003561 | IF (1:200) |
| Antibody | Anti-α-tubulin YL1/2 (Rat monoclonal) | Abcam | Cat#: ab6160, RRID:AB_305328 | IF (1:300) |
| Antibody | Anti-KIF13B (Rabbit polyclonal) | This paper | | WB (1:500) |
| Antibody | Alexa Fluor 488-, 594- and 647-secondaries | Molecular Probes | | IF (1:300) |
| Antibody | IRDye 680LT and 800CW secondaries | Li-Cor Biosciences | | WB (1:15000) |
| Sequence-based reagent | siRNA against luciferase (control) | | | CGTACGCGGA ATACTTCGA |
| Sequence-based reagent | siRNA against Eg5 | This paper | | GAGCCCAGATC AACCTTTA |
| Sequence-based reagent | siRNA against MAP7D1 | *Hooikaas et al., 2019*, PMID:30770434 | | TCATGAAGAGG ACTCGGAA |
| Sequence-based reagent | siRNA against MAP7D3 | *Hooikaas et al., 2019*, PMID:30770434 | | AACCTACATTCG TCTACTGAT |
| Sequence-based reagent | gRNA targeting sequence against KIF13B | This paper | | TGCGGATACGA CCCATGAAC |
| Sequence-based reagent | gRNA targeting sequence against KIF5B | This paper | | CCGATCAAAT GCATAAGGCT |
| Sequence-based reagent | gRNA targeting sequence against KIF1B | This paper | | GCTGGTCTCT CGAGAATTGA |
| Sequence-based reagent | gRNA targeting sequence against KIF1C | This paper | | GCTGGTCTCA CGGGCGTTAA |
| Sequence-based reagent | gRNA targeting sequence against MAP7 | *Hooikaas et al., 2019*; PMID:30770434 | | CGCCCTGCCT CTGCAATTTC |

*Continued on next page*

*Appendix 1—key resources table continued*

| Reagent type (species) or resource | Designation | Source or reference | Identifiers | Additional information |
|---|---|---|---|---|
| Commercial assay or kit | HiPerfect | Qiagen | #301104 | |
| Commercial assay or kit | FuGENE6 | Promega | #E2691 | |
| Chemical compound, drug | Thymidin | Sigma-Aldrich | #T1895 | |
| Chemical compound, drug | Puromycin | InvivoGen | #ant-pr5b | |
| Chemical compound, drug | Biotin | Sigma-Aldrich | #B4639 | |
| Chemical compound, drug | Rapalog | Clontech | AP21967 | |
| Software, algorithm | ImageJ | ImageJ (http://imagej.nih.gov/ij/) | RRID:SCR_003070 | |
| Software, algorithm | FlowJo | FlowJo (https://flowjo.com) | (RRID:SCR_008520) | |
| Software, algorithm | GraphPad Prism | GraphPad Prism (https://graphpad.com) | RRID:SCR_015807 | |
| Software, algorithm | ImageJ detection of molecules plugin | *Chazeau et al., 2016*; PMID:26794511 | | |
| Software, algorithm | ImageJ SOS and SAID plugins | https://imagescience.org/meijering/software/beta/ *Yao et al., 2017*; PMID:28324611 | | |
| Software, algorithm | MATLAB code for track analysis | https://doi.org/10.6084/m9.figshare.c.5177636.v1 | | |
| Software, algorithm | MATLAB simpletracker code | https://www.github.com/tinevez/simpletracker | | |
| Other | Mitotracker Red | Invitrogen | #M7512 | |

