## [Decision Letter]

**Acceptance summary:**

Using a combination of genetic, cell biological, and cutting-edge imaging techniques, this paper defines the individual and cooperative roles of the microtubule motors, kinesin-1 and kinesin-3, in vesicle transport. Although multiple motor populations exist on the same cargo, how distinct motors compete and cooperate to ensure efficient transport are open questions. By precisely analyzing motor position on Rab6 vesicles, this study helps to elucidate the contribution of these different kinesin motor classes to controlling vesicle size, driving regional transport, and opposing dynein in a tug-of-war scenario, thus providing new insights into multimotor cargo transport.

**Decision letter after peer review:**

Thank you for submitting your article "Concerted action of kinesins KIF5B and KIF13B promotes efficient secretory vesicle transport to microtubule plus ends" for consideration by *eLife*. Your article has been reviewed by three peer reviewers, one of whom is a member of our Board of Reviewing Editors, and the evaluation has been overseen by Suzanne Pfeffer as the Senior Editor. The reviewers have opted to remain anonymous.

The reviewers have discussed the reviews with one another and the Reviewing Editor has drafted this decision to help you prepare a revised submission.

Summary:

Serra-Marques, Martin et al. investigate the individual and cooperative roles of specific kinesins in transporting Rab6 secretory vesicles in HeLa cells using CRISPR and live-cell imaging. They find that both KIF5B and KIF13B cooperate in transporting Rab6 vesicles, but Eg5 and other kinesin-3s (KIF1B and KIF1C) are dispensable for Rab6 vesicle transport. They show that both KIF5B and KIF13B localize to these vesicles and coordinate their activities such that KIF5B is the main driver of the cargos on older, MAP7-decorated microtubules, and KIF13B takes over as the main transporter on freshly-polymerized microtubule ends that are largely devoid of MAP7. Interestingly, their data also indicate that KIF5B is important for controlling Rab6 vesicle size, which KIF13B cannot rescue. By analyzing subpixel localization of the motors, they find that the motors localize to the front of the vesicle when driving transport, but upon directional cargo switching, KIF5B localizes to the back of the vesicle when opposing dynein. Overall, this paper provides substantial insight into motor cooperation of cargo transport and clarifies the contribution of these distinct classes of motors during Rab6 vesicle transport.

1) The metrics used to quantify motility are sensitive to tracking errors and uncertainty. The authors quantify the number of runs (Figure 2D, F; Figure 7C) and the average speed (Figure 3A, B, D, E, H). The number of runs is sensitive to linking errors in tracking. A single, long trajectory is often misrepresented as multiple shorter trajectories. These linking errors are sensitive to small differences in the signal-to-noise ratio between experiments and conditions, and the set of tracking parameters used. The average speed is reported only for the long, processive runs (tracks>20 frames, segments<6 frames with velocity vector correlation >0.6). For many vesicular cargoes, these long runs represent <10% of the total motility. In the 4X-KO cells, it is expected there is very little processive motility, yet the average speed is higher than in control cells. Frame-to-frame velocities are often over-estimated due to the tracking uncertainty. Metrics like mean-squared displacement are less sensitive to tracking errors, and the velocity of the processive segments can be determined from the mean-squared displacement (see for example Chugh et al., 2018, Biophys. J.). The authors should also report either the average velocity of the entire run (including pauses), or the fraction of time represented by the processive segments to aid in interpreting the velocity data.

2) The authors show that transient expression of either KIF13B or KIF5B partially rescues Rab6 motility in 4X-KO cells and that knock-out of KIF13B and KIF5B have an additive effect. They also analyze two vesicles where KIF13B and KIF5B co-localize on the same vesicle. The authors conclude that KIF13B and KIF5B cooperate to transport Rab6 vesicles. However, the nature of this cooperation is unclear. Are the motors recruited sequentially to the vesicles, or at the same time? Is there a subset of vesicles enriched for KIF13B and a subset enriched for KIF5B? Is motor recruitment dependent on localization in the cell? These open questions should be addressed in the Discussion.

3) The imaging and tracking of fluorescently-labeled kinesins in cells as shown in Figure 4 is impressive. This is often challenging as kinesin-3 forms bright accumulations at the cell periphery and there is a large soluble pool of motors, making it difficult to image individual vesicles. The authors should provide additional details on how they addressed these challenges. Control experiments to assess crosstalk between fluorescence images would increase confidence in the colocalization results.

4) In Figure 5, it is very interesting that only KIF5B opposes dynein. It would be informative to determine which kinesin was engaged on the Rab6 vesicle before the switch to the retrograde direction. Can the authors analyze the velocity of the run right before the switch to the retrograde direction? If the velocity corresponds with KIF5B (the one example provided seems to show a slow run prior to the switch), this could indicate that KIF5B opposes dynein more actively because KIF5B was the motor that was engaged at the time of the switch. Or if the velocity corresponds with KIF13B, this could indicate that KIF5B becomes specifically engaged upon a direction reversal. In any case, an analysis of the speed distributions before the switch would be provide insight into vesicle movement and motor engagement before the change in direction.

5) In Figure 8G, the tracks for KIF13B-380 motility are difficult to see, which is surprising as KIF13B has been shown to be a superprocessive motor. Is this construct a dimer? If not, do the authors interpret the data as a high binding affinity of the monomer for new microtubules and if so, do they have any speculation on what could be the molecular mechanism? It appears as if KIF13B-380 and EB3 colocalize at the plus ends for a period of time before both are lost but then quickly replenished. Is this common?

6) In general, the reviewers were interested in the localization and velocity of KIF5B vs. KIF13B cargo runs. One reviewer pointed out that in Figure 2E, it seems like about half of the KIF5B events start at or near the Golgi whereas most of the KIF13B events are away from the Golgi? Did the authors find this to be generally true or just apparent in these example images? Furthermore, the results in Figure 8 suggest a potential switch from KIF5B to KIF13B motor engagement upon a change in lattice/MAP7 distribution. Is the velocity of Rab6 vesicles different on central vs. peripheral microtubules? For example, in Figure 4E, are the two examples in different regions of the cell? Do the authors think the intermediate speeds are a result of the motors switching roles? Additional analysis and discussion would strengthen the paper.

---

## [Author Response]

Revisions for this paper:1) The metrics used to quantify motility are sensitive to tracking errors and uncertainty. The authors quantify the number of runs (Figure 2D, F; Figure 7C) and the average speed (Figure 3A, B, D, E, H). The number of runs is sensitive to linking errors in tracking. A single, long trajectory is often misrepresented as multiple shorter trajectories. These linking errors are sensitive to small differences in the signal-to-noise ratio between experiments and conditions, and the set of tracking parameters used. The average speed is reported only for the long, processive runs (tracks>20 frames, segments<6 frames with velocity vector correlation >0.6). For many vesicular cargoes, these long runs represent <10% of the total motility. In the 4X-KO cells, it is expected there is very little processive motility, yet the average speed is higher than in control cells. Frame-to-frame velocities are often over-estimated due to the tracking uncertainty. Metrics like mean-squared displacement are less sensitive to tracking errors, and the velocity of the processive segments can be determined from the mean-squared displacement (see for example Chugh et al., 2018, Biophys. J.). The authors should also report either the average velocity of the entire run (including pauses), or the fraction of time represented by the processive segments to aid in interpreting the velocity data.

Two stages of the described tracking and data processing are responsible for the extraction of processive runs: the “linking” method used during the tracking, and the “trajectory segmentation” method, applied to the obtained tracks. The detection and linking of vesicles have been performed using our previously published tracking method (Chenouard et al., 2014, Nature Methods, PMID: 24441936). Our linking method uses multi-frame data association, taking into account detections from four subsequent image frames in order to extend and create a trajectory at any given time. This allows for dealing with temporal disappearance of particles (missing detections) for 1-2 frames and avoiding creation of breaks in longer trajectories. The method is robust to noise, spurious and missing detections and had been fully evaluated in the aforementioned paper (Chenouard et al., 2014) showing excellent performance compared to other tracking methods.

Having the trajectories describing the behavior of each particle, the track segmentation method had been applied to split each trajectory into a sequence of smaller parts (tracklets) describing processive runs and pieces of undirected (diffusive) motion. The algorithm that we used was validated earlier on an artificial dataset (please see Supplementary Figure 2E in Katrukha et al., 2017). The chosen parameters were in the range where the algorithm provided less than 10% of false positives. Since the quantified and reported changes in the number of runs are six-fold (Figure 2D, F), we are quite certain that this estimated error (inherent to all automatic image analysis methods) does not affect our conclusions. Moreover, it is consistent with visual observations and manual analysis of representative videos.

Further, we agree that frame-to-frame velocities are often somewhat over-estimated due to the tracking uncertainty. We are aware of such overestimation which is very difficult to avoid. In our case, we estimated (using a Monte Carlo simulation) that such overestimation will positively bias the average not more than 3-6%. Since we focus not on the absolute values of velocities, but rather on the comparison between different conditions, such biasing will be present in all estimates of average velocity and will not affect the presented conclusions.

The usage of mean square displacement (MSD) to analyze trajectories containing both periods of processive runs and diffusive motion is confusing, since it represents average value over whole trajectories, resulting in the MSD slope which is in the range of 1.5 (i.e. between 1, diffusive and 2, processive; please see Figure 2C in Katrukha et al., 2017). Therefore, initial segmentation of trajectories is necessary, as it was performed in the paper by Chugh et al. (Chugh et al., 2018, Biophysical Journal, PMID: 30021112; please see Figure 2E in that paper), suggested by the reviewer. In this paper the authors used an SCI algorithm, which is very similar to our analysis, relying on temporal correlations of velocities. Indeed, MSD analysis of only processive segments is less sensitive to tracking errors, but it reports an average velocity of the whole population of runs. This method is well suited if one would expect monodisperse velocity distribution (the case in Chugh et al., where single motor trajectories are analyzed). If there are subpopulations with different speeds (as we observed for Rab6 by manual kymograph analysis), this information will be averaged out. Therefore, we used histogram/distribution representations for our speed data, which in our opinion represents these data better.

Finally, we fully agree with the reviewers that the fractions of processive/diffusive motion should be reported. In the revised version, we have added new plots (new Figure 2G-I, Figure 2—figure supplement 2G) illustrating these data for different conditions. Our data fully support the reviewer’s statement that processive runs represent less than 10% of total vesicle motility (new Figure 2G). As could be expected, the total time vesicles spent in processive motion and the percentage of trajectories containing processive runs strongly depended on the presence of the motors (new Figure 2H, I). However, within trajectories that did have processive segments, the percentage of processive movement was similar (new Figure 2I).

We note that while our analysis is geared towards identification and characterization of processive runs (which was verified manually), analysis of diffusive movements poses additional challenges and is even more sensitive to linking errors. Therefore, we do not make any strong quantitative conclusions about the exact percentage and the properties of diffusive vesicle movements, and their detailed studies will require additional analytic efforts.

In addition, we made publicly available all raw and segmented trajectories, along with corresponding code, by sharing them in Figshare repository. It opens an opportunity for interested researchers to inspect and re-analyze our findings.

2) The authors show that transient expression of either KIF13B or KIF5B partially rescues Rab6 motility in 4X-KO cells and that knock-out of KIF13B and KIF5B have an additive effect. They also analyze two vesicles where KIF13B and KIF5B co-localize on the same vesicle. The authors conclude that KIF13B and KIF5B cooperate to transport Rab6 vesicles. However, the nature of this cooperation is unclear. Are the motors recruited sequentially to the vesicles, or at the same time? Is there a subset of vesicles enriched for KIF13B and a subset enriched for KIF5B? Is motor recruitment dependent on localization in the cell? These open questions should be addressed in the Discussion.

Unfortunately, only fluorescent motors and not the endogenous ones can be detected on vesicles, so we cannot make any strong statements on this issue. Since KIF13B can compensate for the absence of KIF5B, it can be recruited to the vesicle when it emerges from the Golgi apparatus. However, in normal cells, KIF5B likely plays a more prominent role in pulling the vesicles from the Golgi, as Rab6 vesicles generated in the presence of KIF5B are larger (Figure 5I). We show in Figure 1G, H that KIF13B does not exchange on the vesicle and stays on the vesicle until it fuses with the plasma membrane. These data suggest that once recruited, KIF13B stays bound to the vesicle. Obtaining such data for KIF5B is more problematic because fewer copies of this motor are typically recruited to the vesicle (Figure 4B) and its signal is therefore weaker. Further research with endogenously tagged motors and highly sensitive imaging approaches will be needed to address the important open questions raised by the reviewer. We have added these points to the second and fourth paragraphs of the Discussion.

3) The imaging and tracking of fluorescently-labeled kinesins in cells as shown in Figure 4 is impressive. This is often challenging as kinesin-3 forms bright accumulations at the cell periphery and there is a large soluble pool of motors, making it difficult to image individual vesicles. The authors should provide additional details on how they addressed these challenges. Control experiments to assess crosstalk between fluorescence images would increase confidence in the colocalization results.

Imaging of vesicle motility was performed using TIRF microscopy focusing on regions where no strong motor accumulation was observed. We have little cross-talk between red and green channels, but channel cross talk in the three-color images shown in Figure 4E was indeed a potential concern. To address this potential issue, we performed the appropriate controls and added a new figure (Figure 4—figure supplement 1). We conclude that we can reliably simultaneously detect blue, green and red channels without significant cross-talk on our microscope setup.

4) In Figure 5, it is very interesting that only KIF5B opposes dynein. It would be informative to determine which kinesin was engaged on the Rab6 vesicle before the switch to the retrograde direction. Can the authors analyze the velocity of the run right before the switch to the retrograde direction? If the velocity corresponds with KIF5B (the one example provided seems to show a slow run prior to the switch), this could indicate that KIF5B opposes dynein more actively because KIF5B was the motor that was engaged at the time of the switch. Or if the velocity corresponds with KIF13B, this could indicate that KIF5B becomes specifically engaged upon a direction reversal. In any case, an analysis of the speed distributions before the switch would be provide insight into vesicle movement and motor engagement before the change in direction.

Directional switching was only analyzed in rescue experiments, where the vesicles were driven by either KIF5B alone or by KIF13B alone, and the speeds of vesicles were representative of these motors (see Author response image 1). The number of vesicle runs where two motors were detected simultaneously (KIF5B vs. KIF13B in Figure 5G, H, J) were significantly lower, and therefore, unfortunately we could not perform the analysis of their directional switching with sufficient statistical power.

5) In Figure 8G, the tracks for KIF13B-380 motility are difficult to see, which is surprising as KIF13B has been shown to be a superprocessive motor. Is this construct a dimer? If not, do the authors interpret the data as a high binding affinity of the monomer for new microtubules and if so, do they have any speculation on what could be the molecular mechanism? It appears as if KIF13B-380 and EB3 colocalize at the plus ends for a period of time before both are lost but then quickly replenished. Is this common?

KIF13B-380 construct used here contains a leucine zipper from GCN4 and is therefore dimeric. We have now indicated this more clearly in the Results subsection “Combination of KIF5B and KIF13B allows Rab6 vesicles to reach growing MT plus ends”. KIF13B-380 does show processive motility, although this is difficult to see close to the outermost microtubule tip as the motor tends to accumulate there. We now provide a kymograph in Figure 8G where the processive motility of KIF13B-380 is clearer.

6) In general, the reviewers were interested in the localization and velocity of KIF5B vs. KIF13B cargo runs. One reviewer pointed out that in Figure 2E, it seems like about half of the KIF5B events start at or near the Golgi whereas most of the KIF13B events are away from the Golgi? Did the authors find this to be generally true or just apparent in these example images?

We sincerely apologize for the misunderstanding here. To automatically track the vesicles, we had to manually exclude the Golgi area. Moreover, only processive and not complete tracks are shown. Therefore, no conclusions can be made from these data on the vesicle exit from the Golgi. We now indicated this clearly in the Results (subsection “KIF5B and KIF13B are the main drivers of Rab6 vesicle transport in HeLa cells”, second paragraph) and Discussion (fourth paragraph) and included more representative images in Figure 2E.

Furthermore, the results in Figure 8 suggest a potential switch from KIF5B to KIF13B motor engagement upon a change in lattice/MAP7 distribution. Is the velocity of Rab6 vesicles different on central vs. peripheral microtubules? For example, in Figure 4E, are the two examples in different regions of the cell?

This is indeed a very interesting point and we have considered it carefully. As can be seen in Figure 8B (grey curve), vesicle speed remains relatively constant along the cell radius in control HeLa cells. We note, however, that our previous work has shown that in these cells microtubules are quite stable even at the cell periphery, due to the high activity of the CLASP-containing cortical microtubule stabilization complex (Mimori-Kiyosue et al., 2005, Journal of Cell Biology, PMID: 15631994; van der Vaart et al., 2013, Developmental Cell, PMID: 24120883). We therefore hypothesized that changes in vesicle speed distribution along the cell radius would be more obvious in cells with highly dynamic microtubule networks and performed a preliminary experiment in MRC5 human lung fibroblasts, which have a very sparse and dynamic microtubule cytoskeleton (Splinter et al., 2012). As shown in Author response image 2, we indeed found that vesicles move faster at the cell periphery. Even though these data are suggestive, characterization of this additional cell model goes beyond the scope of the current study, and we prefer not to include them in the manuscript. However, since this paper is part of preprint review and our responses will be published on bioRxiv, the data will be available to the community.

**Author response image 2. respfig2:** 

In Figure 4E, the two examples are from different cells, and both recorded at the cell periphery. The difference in vesicle speeds reflects general speed variability.

Do the authors think the intermediate speeds are a result of the motors switching roles? Additional analysis and discussion would strengthen the paper.

Presence of intermediate speeds of cargos driven by multiple motors of two types is most clear in Figure 3F-H, where multiple and different ratios of KIF5B and KIF13B motors are recruited to peroxisomes. As can be seen in Figure 3G, the kymographs in these conditions are “smooth” and no evidence of motor switching can be detected at this spatiotemporal resolution. On the other hand, it has been previously beautifully shown by the Verhey lab that when artificial cargos are driven by just two motor molecules of different nature, switching does occur (Norris et al., 2014). This point is emphasized in the third paragraph of the subsection “KIF5B and KIF13B have different velocities and set the speed range of Rab6 vesicles”. These data suggest that motors working in teams show different properties, and more detailed biophysical analysis will be needed to understand them.